JCB Journal of Cell Biology

## TOOLS

# AI-directed voxel extraction and volume EM identify intrusions as sites of mitochondrial contact

Benjamin S. Padman[1,4], Runa S.J. Lindblom[2,3,4], and Michael Lazarou[1,2,3,4]

Membrane contact sites (MCSs) establish organelle interactomes in cells to enable communication and exchange of materials. Volume EM (vEM) is ideally suited for MCS analyses, but semantic segmentation of large vEM datasets remains challenging. Recent adoption of artificial intelligence (AI) for segmentation has greatly enhanced our analysis capabilities. However, we show that organelle boundaries, which are important for defining MCS, are the least confident predictions made by AI. We outline a segmentation strategy termed AI-directed voxel extraction (AIVE), which refines segmentation results and boundary predictions derived from any AI-based method by combining those results with electron signal values. We demonstrate the precision conferred by AIVE by applying it to the quantitative analysis of organelle interactomes from multiple FIB-SEM datasets. Through AIVE, we discover a previously unknown category of mitochondrial contact that we term the mitochondrial intrusion. We hypothesize that intrusions serve as anchors that stabilize MCS and promote organelle communication.

## Introduction

Organelles allow eukaryotic cells to compartmentalize cellular pathways that would normally be incompatible in a shared environment. However, compartmentalization also isolates key processes that depend on one another. To link isolated pathways, cells dynamically regulate interactions between organelles via physical points of contact called membrane contact sites (MCSs) (Voeltz et al., 2024) that mediate inter-organelle communication, material exchange, tethering, and structural support (Scorrano et al., 2019). For example, cardiolipin synthesis at the inner mitochondrial membrane requires the precursor lipid phosphatidic acid, which is synthesized by the ER (Flis and Daum, 2013); without adequate MCSs between mitochondria and the ER, cardiolipin synthesis would halt. MCSs are now widely accepted to be essential for cellular homeostasis (Schrader et al., 2015), and MCSs have been reported between almost every conceivable combination of organelles (Coleman, 2019; Shai et al., 2016; Venditti et al., 2020; Wang et al., 2022). Collectively, the interaction network between these various organelles is referred to as the "organelle interactome" (Valm et al., 2017).

Meaningful analysis of the organelle interactome poses a major technical challenge because MCSs represent small points of contact between comparatively large structures. Optical microscopy is suitable for imaging large interacting structures, but the separation distances of an MCS at 10–35 nm (Jing et al., 2019) are smaller than the resolvable limits of light. Biochemical methods also have limitations, since most sample preparation methods typically remove structural and spatial context from the MCS prior to analysis (Huang et al., 2020). This is why volume EM (vEM) techniques are frequently referred to as the "gold standard" for studying MCS structure (Jing et al., 2019).

vEM methods based on scanning EM (SEM), such as focused ion beam SEM (FIB-SEM) or serial block face SEM, enable the ultrastructural visualization of membrane structures within cellular-scaled volumes. However, analysis of large vEM datasets poses a major challenge since the structures detected require semantic segmentation (hereafter segmentation) and object classification before they can be quantitatively analyzed, and this process has historically relied on the manual labeling of each organelle membrane by hand to define their boundaries. This reliance on human input represents a major bottleneck in the throughput of vEM methods, and the traditional approaches to image segmentation can no longer keep pace with the rate of data acquisition (Peng et al., 2018), which increases every year (Xu et al., 2020). Given these challenges, it can be argued that the rapid adoption of machine learning– and

..........................................................................................................................................................................................................................
[1]Department of Biochemistry and Molecular Biology, Biomedicine Discovery Institute, Monash, Australia; [2]Walter and Eliza Hall Institute of Medical Research, Parkville, Australia; [3]Department of Medical Biology, University of Melbourne, Melbourne, Australia; [4]Aligning Science Across Parkinson's (ASAP) Collaborative Research Network, Chevy Chase, MD, USA.

Correspondence to Michael Lazarou: lazarou.m@wehi.edu.au; Benjamin S. Padman: benjamin.padman@uwa.edu.au

B.S. Padman's current affiliation is The Kids Institute, Perth Children's Hospital, Nedlands, Australia.

deep learning– based approaches for vEM was inevitable (Gallusser et al., 2023; Liu et al., 2022). Artificial intelligence (AI)-based image segmentation approaches are now widely accepted in the field of vEM, but they are not without their own potential flaws when it comes to organelle boundaries. Segmentation aims to define where an object is, but it also aims to define where an object is not. The transition between the presence or absence of an object defines its bounding surface in a vEM dataset. The challenge arises when organelle surfaces segmented via AI-based approaches represent locations where a trained model was equally certain about the presence and absence of that object, and this can become problematic for MCS analyses.

Here, we demonstrate that the least certain model predictions made via AI-based image segmentation occur predominantly at the bounding surface of an object, which is the most important location for MCS analyses. This boundary uncertainty causes variable results between AI models and directly impacts the reliability and accuracy of ultrastructural measurements. We outline and benchmark a strategy called AI-directed voxel extraction (AIVE) designed to negate this limitation. Through a series of benchmarking experiments, we demonstrate AIVE's capacity to reduce the variability between different AI models and confer greater consistency in MCS analyses in cultured cells and tissues alike. We conduct a comprehensive analysis of FIB-SEM datasets acquired from HeLa cells, revealing the existence of a previously undescribed morphological category of mitochondrial MCSs that we term mitochondrial intrusions. The benefits of mitochondrial intrusions include increased surface area of membrane contacts between organelles that may serve as an anchor to enhance cross talk efficiency.

## Results

### AI-based approaches for the segmentation of vEM data

AI-based image segmentation uses information learned from training data to create statistical or deductive frameworks (termed a model) so that predictions can be made about new unseen data. Given the large datasets that can be generated using FIB-SEM imaging of cells, there are clear advantages in using AI-based approaches to automate membrane segmentation (Nguyen et al., 2021). To demonstrate, we used a test dataset of ~3.5 μm³ in volume (dimensions: X, 2.4 μm [740 px]; Y, 1.2 μm [370 px]; Z, 1.2 μm [121 slices]), which originates from a larger 655=μm³ dataset that will be discussed later in greater detail (Figs. 4, 5, 6, 7, and 8). The test dataset was used to train a random forest (RF) model to identify cellular membranes detected via FIB-SEM imaging (Fig. 1, A and B). This trained model generated probability (P) values indicating the likelihood that a membrane is present at each voxel in the dataset (Fig. 1 C). To complete the segmentation of these membranes (Fig. 1, D and E), the scalar P values were binarized by assigning a probability threshold (Fig. 1 D) or an isosurface value (Fig. 1 E) to generate a 3D result. The decision threshold is a critical parameter in AI-based image segmentation because it defines the boundary of each segmented object, thereby defining an object's volume, surface area, and contact with other objects. When selecting a

threshold value, a midpoint P value of 50% appears to be a logical choice (Fig. 1, D and E), but 50% probability values represent where the model was equally certain about the presence and absence of an object, making it the least certain prediction. Deep learning methods like convolutional neural networks (CNNs) do not use a pre-chosen decision threshold, but the output layer of a CNN learns parameters akin to a threshold to generate a binary result (Richard and Lippmann, 1991; Rumelhart et al., 1986). Regardless of the methods used to predict and binarize results, the decision thresholds control the ultimate conclusions drawn by a model and, by extension, the boundary of each object.

When presented with the same data, different AI algorithms make different predictions about the boundaries of an object, and these differences can affect interpretations of organelle ultrastructure and membrane contacts (Wang et al., 2020). To demonstrate this, we trained six machine learning models based on different algorithms of varying complexity to detect membranes in the same test dataset from Fig. 1 B. All six models were trained in the detection of membranes using the exact same training samples and feature data before application to the same test dataset (Fig. 1 F), and therefore any variability between results would be due to their algorithmic differences. Each model was capable of membrane detection (Fig. 1 F), but key differences existed between their results. By visualizing the differences in predicted values (ΔV) from each model (Fig. 1 G; ΔV), it can be demonstrated that most of the predictions agree between models, but the disagreements that do occur are almost exclusively localized to the membrane boundaries (Fig. 1 G). This uncertainty about the position of a membrane boundary would negatively impact the reliability of any organelle contact analysis, since it confounds the measurement of membrane separation distances.

### AIVE: A tool for accurate automated segmentation of cellular membranes

To improve the accuracy of membrane boundaries obtained from AI segmentation methods and to generate greater consensus across different AI algorithms, we developed an approach termed AIVE (Lee et al., 2024; Nguyen et al., 2021). AIVE combines the automated segmentation outputs of an AI model with the ground truth of the vEM data. Applying AIVE involves two major steps: (1) Segmentation of cellular membranes with an AI (as in Fig. 1), and then (2) multiplication of the AI membrane prediction with the original EM data to extract only those voxels belonging to a membrane. This multiplication is conducted using the raw probabilistic model predictions, although we note that earlier iterations of AIVE used binary segmented data for this stage (Nguyen et al., 2021). The benefit of AIVE for membrane boundary detection is that the regions of uncertain AI prediction are accounted for by the intensity of signals in the EM data. For example, at a boundary where the AI is uncertain, if the EM voxels have little to no intensity, then the AI prediction of the uncertain region will not contribute to the output (Fig. S1 A). Unlike methods that use EM signals to augment AI performance or training (Brion et al., 2021; Liu and Ji, 2021), the core stages of AIVE occur downstream of AI-based image segmentation without influencing the training or behavior of the AI model. This is

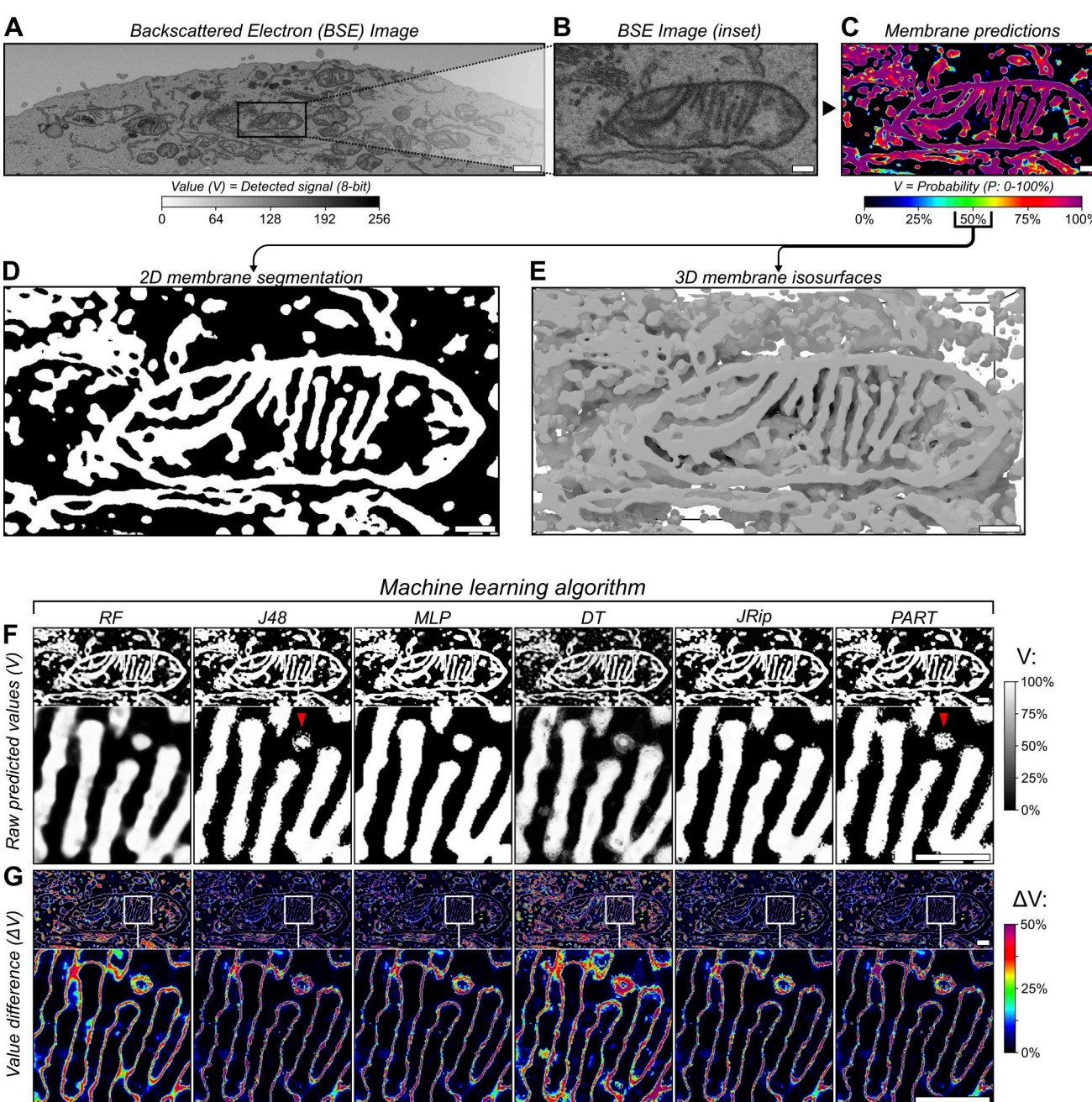

Figure 1. **Variable AI predictions predominantly occur at object boundaries during segmentation. (A–E)** 3D test data acquired via FIB-SEM imaging, showing an (A) overview and (B) inset 2D slice from the dataset with (C) corresponding predictions generated by a random forest classifier trained to detect membranes, alongside (D) 2D binary segmentation and (E) 3D surfaces based on the membrane predictions. **(F)** 2D slices depicting raw membrane predictions for the test dataset (from Fig. 1 B) with values (V) shown as percentages of their total dynamic range (0–1), made using six different machine learning algorithms (also provided as 3D renders in Fig. S2 A); RF, J48, MLP (multilayer perceptron), DT (decision table), JRip, and PART (projective adaptive resonance theory). **(G)** Visual depiction of the absolute difference in voxel values (ΔV) between each prediction from F, showing the average value difference for each two-way value comparison as a percentage of the total dynamic range (0–1). Scale bars; A, 1,000 nm; B–G, 200 nm.

an important distinction because it means that AIVE can be applied to any existing data without requiring adjustments to preexisting trained models for that data.

To demonstrate the overall AIVE approach, the sample EM data and preexisting AI predictions from Fig. 1 were applied to the AIVE workflow. First, the AI prediction was smoothened by applying a 3D Gaussian blur to reduce variability between

predictions (Fig. S1 C). The EM data were processed using contrast limited adaptive histogram equalization (CLAHE) (Fig. 2 C) to normalize differences in image contrast that can occur between different data acquisitions and median blurred to reduce signal shot noise. In the final step, the voxel values from the AI predictions and the raw EM data (Fig. 2, B and C) were multiplied to generate the AIVE output (Fig. 2 D). By multiplying voxel

values from two independent dynamic ranges (Fig. S1 A), any input voxels with a value of zero were automatically excluded from the output, whereas electron signals predicted by the model to represent membranes were retained for segmentation (Fig. 2 D and Fig. S1 A). Importantly, this strategy is not limited to membranes, since any cellular content that generates an electron signal can be isolated (Fig. S1 B). Like any other segmentation technique, a threshold value is required to generate 3D surfaces from the AIVE dataset (Fig. S2, A–C). It is important to note that values near the center of the output dynamic range (50%; 8-bit value of 128) are not equivalent to the 50% probability threshold value discussed earlier (Fig. 1 D). This is because AIVE results are the product of multiplying two independent dynamic ranges (Fig. S1 A), so the center of both dynamic ranges is in fact 25% (50% × 50% = 25%; 8-bit value of 64). The premise of AIVE is therefore akin to a signal masking strategy, in which electron signals detected during EM are used for the segmentation of object boundaries instead of using AI predictions alone (Fig. 1 G). Doing so improves the accuracy, precision, and consistency of membrane segmentation, as we will demonstrate through a series of benchmarking experiments.

## Benchmarking AIVE

### AIVE enables outputs to be compared between different AI algorithms

We had observed large discrepancies between membrane boundary positions segmented by different AI models in the same EM dataset (Fig. 1 F). To benchmark AIVE and demonstrate its benefits, the same AI model predictions shown in Fig. 1 F were applied to the AIVE workflow (Fig. 2 E). Upon application of AIVE, the difference in values (ΔV) between each model greatly reduced (compare Fig. 1, F and G to Fig. 2, E and F). Quantitation of the ΔV per image slice also demonstrated that the variability between results can be more than halved by AIVE, though we note that some of this improvement was due to the initial blurring of the predictions (Fig. S1 C). We also note that the principles of AIVE are applicable to 2D imaging methods such as transmission EM (TEM). Similar to the 3D AIVE analyses (Figs. 1 and 2), 2D application of AIVE reduced the variability between results generated by different AI models (Fig. S3).

Next, we assessed how the differences between boundaries from each model would affect 3D spatial measurements. This was done by selecting 30 random points near membranes (distance probes; Fig. 2 H; see also Video 1), then measuring the distance between each point and its nearest membrane (Fig. S2 D). By measuring the range (Fig. 2 I) and relative absolute deviation (Fig. 2 J) of distance values recorded between models for each probe, the reliability of measurements under each condition can be quantified. By both measures (Fig. 2, I and J), AIVE processing greatly improved the consistency of measurements between models, outperforming the AI predictions alone with or without the application of a Gaussian blur. Together, these analyses show that AIVE enables greater consensus between AI models at regions of uncertainty (membrane boundaries)

and that AIVE can be applied with any AI model chosen by a user.

### AIVE improves the accuracy and precision of organelle classification

The accuracy of organelle membrane segmentation is fundamentally important for the analysis of organelle interactions. The AIVE framework can easily integrate organelle classification prior to the voxel extraction (VE) step. This simply requires the preparation of binary labels that map the identity of organelles. For example, if a mitochondrion of interest were present in the dataset, its identity can be defined by drawing a label around that approximate region (Fig. 3 A). While this process may seem identical to conventional classification, AIVE differs in that it does not use these classification labels to define the boundaries of any organelles. Instead, the class labels are used to mask regions of the AI predictions and merely identify them as belonging to a specific organelle (Fig. 3, B and C). The resulting data are then processed via AIVE, resulting in segmentation of membranes belonging to a specific organelle, which in the sample case is a mitochondrion (Fig. 3, D and E).

Class labels can be assigned manually by a human, automatically by AI, or any combination of those two strategies via "human-in-the-loop" type approaches (Budd et al., 2021). This is a valuable feature of AIVE, as it allows human experts to contribute their domain knowledge when an AI encounters structures that it cannot classify, without requiring the training of an entirely new classification model. To demonstrate the flexible classification requirements of AIVE, we assessed the classification labels generated by a human analyst and two U-Net CNNs in the detection of one mitochondrion (Fig. 3 F) from the test dataset in Fig. 1 A. Classification by a human analyst was conducted manually on two consecutive days, while the two U-nets differed in their randomization seed during training. Mitochondrial labeling was generally similar between the classification approaches (Fig. 3 F), but the human classifications tended to extend further than necessary beyond the perimeter of mitochondria, whereas the U-Nets occasionally appeared to miss portions of the mitochondrion (Fig. 3 F; arrowheads).

To benchmark the benefits of AIVE, the binary class labels from Fig. 3 F were used for organelle segmentation using four different approaches: (1) Traditional instance segmentation, in which binary class labels are used to define object boundaries (Fig. 3 G); (2) AI extraction (AIE), where the class labels are used to mask the AI predictions without VE (Fig. 3 H); (3) direct VE, which uses the class labels to mask the normalized EM data without using AI predictions (Fig. 3 I); and (4) AIVE (Fig. 3 J).

To understand the reconstruction results, we must first address the characteristics and limitations of conventional 3D segmentation. Specifically, 3D surfaces generated using binary segmentation alone (Fig. 3 G) demonstrate a noteworthy artifact called "terracing" (Fig. 3 G; see 3D insets), which presents as artificial plateaus (Gibson, 1998). The terracing is caused by abrupt (binary) changes in voxel value, which affect the local shape since contours can only be represented by a limited pool of surface topologies ($2^8$; 256) (Lorensen and Cline, 1987), of which only 15 are unique (Kim et al., 2017). The rigid pool of surface topologies imposes a limit on the smallest nonzero distance that

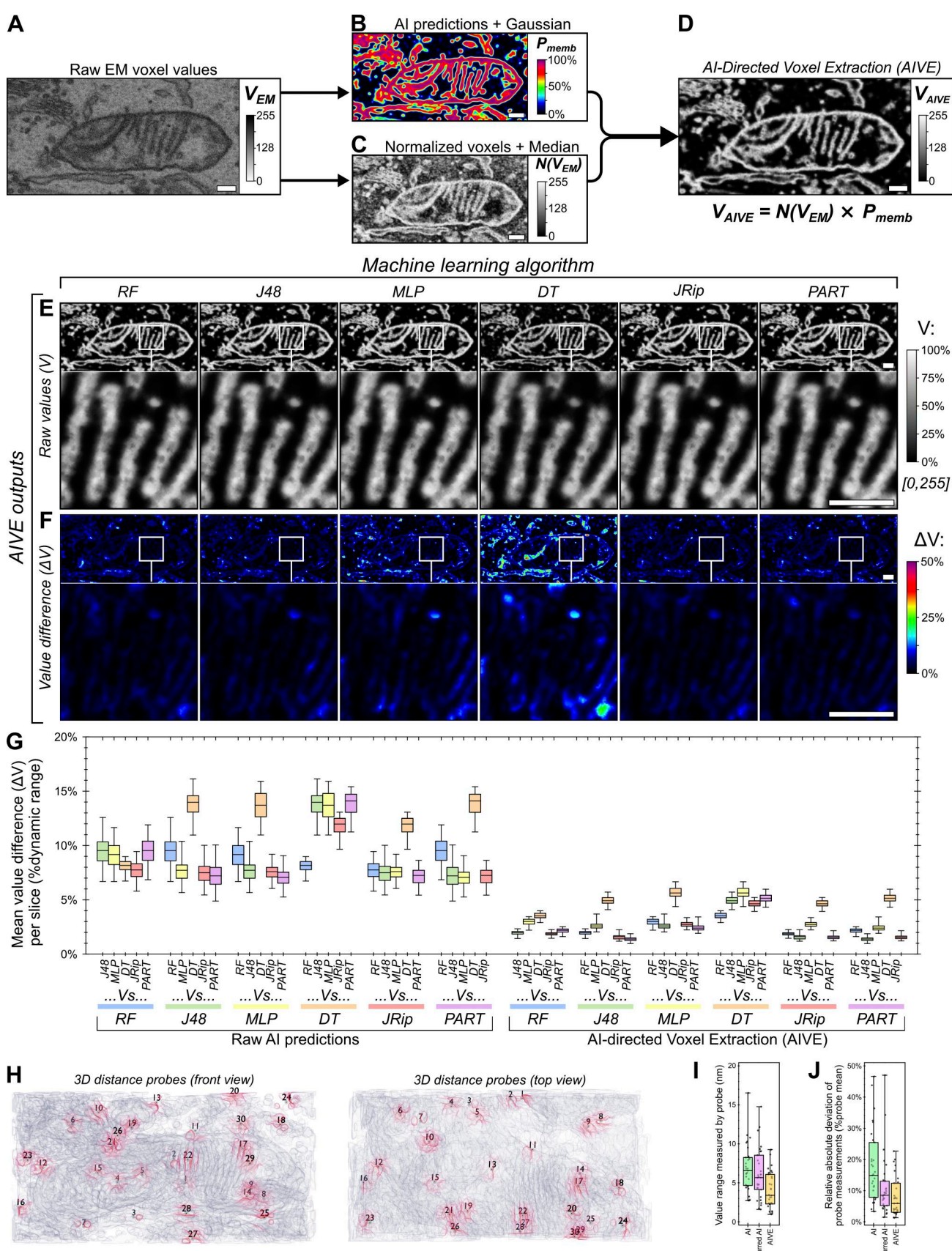

Figure 2. **AIVE minimizes the influence of variable AI predictions. (A–D)** Flow chart depicting the technical basis of AIVE, in which (A) raw 3D data from the FIB-SEM are used to (B) train and apply a classifier model to generate probability maps depicting the location of cellular membranes (dynamic range, 0–1), while

(C) intensity values from the original dataset are normalized via CLAHE (dynamic range, 0–255); (D) values from B are then multiplied by values from C (see also Fig. S1 A), which isolates the voxel values belonging to a membrane to generate raw AIVE data (application of AIVE to other classes of signal provided in Fig. S1 B). **(E and F)** 2D slices depicting raw AIVE data generated using each algorithm shown in Fig. 1 F (3D renders of the same data provided in Fig. S2 C), and (F) averaged ΔV values for each two-way comparison between one AIVE output and each of the other outputs (both shown as a percentage of 8-bit dynamic range; 0–255). **(G)** Quantification of the average ΔV per slice for each two-way comparison made between the model predictions (Fig. 1, F and G) and between AIVE outputs (E and F) generated using those same model predictions (quantitation and 3D renders of blurred AI predictions in Fig. S1 C and Fig. S2 B, respectively). **(H)** 3D rendering of membrane surfaces from the test dataset (from Fig. 1 E) viewed from the front and above, indicating the positions of the static 3D probes used to benchmark membrane distance measurements (membranes nearest to each 3D probe are indicated by red shading; see Video 1 for rotation animation depicting probe positions). **(I and J)** Quantification of the distance between each 3D probe (shown in H) and the nearest membranes detected by the six models shown in Fig. 1 F (raw distance measurements provided in Fig. S2 A), after segmentation of the raw predictions, blurred AI predictions, or AIVE outputs, showing (I) the range (min–max) for values reported by each probe between models and (J) the relative absolute deviation in measurements from each probe. Plot markers in I and J indicate value for individual 3D probes. The interquartile range is indicated by the box, the median is indicated by the horizontal lines, and the minimum and maximum are indicated by whiskers in G, I, and J. Scale bars; A–F, 200 nm. MLP, multilayer perceptron; DT, decision table; PART, projective adaptive resonance theory.

can be measured between any two objects, which would be 2.3 nm for our data (see Fig. S1, G–I). In contrast, the additional information present within 8-bit scalar datasets, such as those generated by AIE, VE, and AIVE (Fig. 3, H–J), allows surface vertices to be positioned at any point between voxel centers, theoretically allowing for >1.8 × 10$^{19}$ (256$^8$) different unique polygon configurations. These polygon configurations can also adopt fractional angles that have spatial anisotropy with the voxel grid (Kim et al., 2017), thus removing the limit on the smallest measurements that can be made between objects. Therefore, by moving from binary labels to scalar values, we can generate a more natural contour from our voxel grid, as shown by the surface details shown in Fig. 3, H–J compared with Fig. 3 G.

Next, we benchmarked AIVE against VE and AIE. VE prevents terracing (Fig. 3 H), but the scalar data also include irrelevant electron signals. Similarly, AIE reduces terracing (AIE; Fig. 4 I), but the disadvantages of relying on AI predictions alone for membrane boundaries have already been outlined (Figs. 1 and 2). To directly show how the different approaches influence the reliability of contact site analyses, we quantified the variability of distance measurements between classification labels for each of the scalar 3D datasets (Fig. 3, H–J). We conducted 3D measurements (Fig. 3 K) using a new set of 3D distance probes nearer to a mitochondrion (Fig. S1 D; see also Video 2), since the original 3D probe locations had no relation to a specific organelle (Fig. 2 H). As demonstrated by the range (Fig. 3 L) and relative absolute deviation (Fig. 3 M) of distance values recorded by each 3D probe, the distance measurements recorded for AIVE datasets were very consistent between each of the differing class labels as opposed to VE and AIE (Fig. 3 F). Therefore, AIVE substantially improves the precision and consistency of 3D spatial measurements in vEM datasets.

We next conducted a benchmarking experiment to compare the accuracy of measurements generated via AIVE to a known standard point of reference. Nuclear pore complexes (NPCs) are large protein channels in the nuclear envelope that mediate cargo transport between the interior and exterior of the nucleus. NPC structure is highly conserved among eukaryotes, which is why they are often used as a reference standard for benchmarking microscopy methods (Thevathasan et al., 2019). The test dataset used in Figs. 1, 2, and 3 originated from a larger cellular volume that also contained a portion of nuclear envelope

(Fig. 4 B). Using the manual classification strategy described in Fig. 3, A–J, we processed a portion of the nuclear envelope containing 18 individual nuclear pores (Fig. S1 E) via VE (Fig. 3 N), AIE (Fig. 3 O), or AIVE (Fig. 3 P). The same 18 pores were isolated from each processed dataset and aligned using the exact same transforms (see Materials and methods for further detail; Fig. S1 F) to compare each processing method (Fig. 3, Q–T) against recent NPC structures (Schuller et al., 2021; Zimmerli et al., 2021). Data processed by VE alone (Fig. 3 N) rarely yielded a detectable pore because they were often occluded by irrelevant electron signals (Fig. 3 Q). In contrast, central transport channels were consistently detected in the data processed by AIE alone (Fig. 3 O) or AIVE (Fig. 3 P), but the average diameter of those pores differed between processing methods (Fig. 3 Q): 42.9 nm for AIE alone compared with 52.4 nm for AIVE. We related these pore measurements to reported values from two independent studies from 2021 (arrowheads in Fig. 3 Q), which used cryo-electron tomography to resolve NPC structures in cellulo (Schuller et al., 2021; Zimmerli et al., 2021). Schuller et al. (2021) reported that human NPCs have a central channel diameter of 57 nm (Fig. 3 Q; black arrowhead) (Schuller et al., 2021). The AIE processed data differ from this value by 14.1 nm, whereas the AIVE results differed by only 4.6 nm. The study by Zimmerli et al. (2021) was not conducted in human cells (Zimmerli et al., 2021), but we include their findings because they provide valuable context for the distribution of our measurements (Fig. 3 Q; value range indicated by white arrowheads). Unlike data processed using VE or AIE alone, the majority of the AIVE-processed nuclear pores had a diameter within the published reported range. Renderings of the median averaged pores from each processing method are also provided (Fig. 3, R–T; shown in purple) for visual comparison with the human NPC structure published by Schuller et al. (2021) (Fig. 3, R–T; shown in orange [Schuller et al., 2021]). Collectively, our benchmark experiments show that AIVE processing substantially improves both the precision and accuracy of spatial measurements in vEM datasets.

## Mapping organelle interactomes using AIVE
### Compositional analysis of cellular membranes via AIVE
Having outlined the technical basis and advantages of AIVE, we next demonstrated some of the biological insights AIVE can yield when applied to large-volume datasets. We aimed to characterize

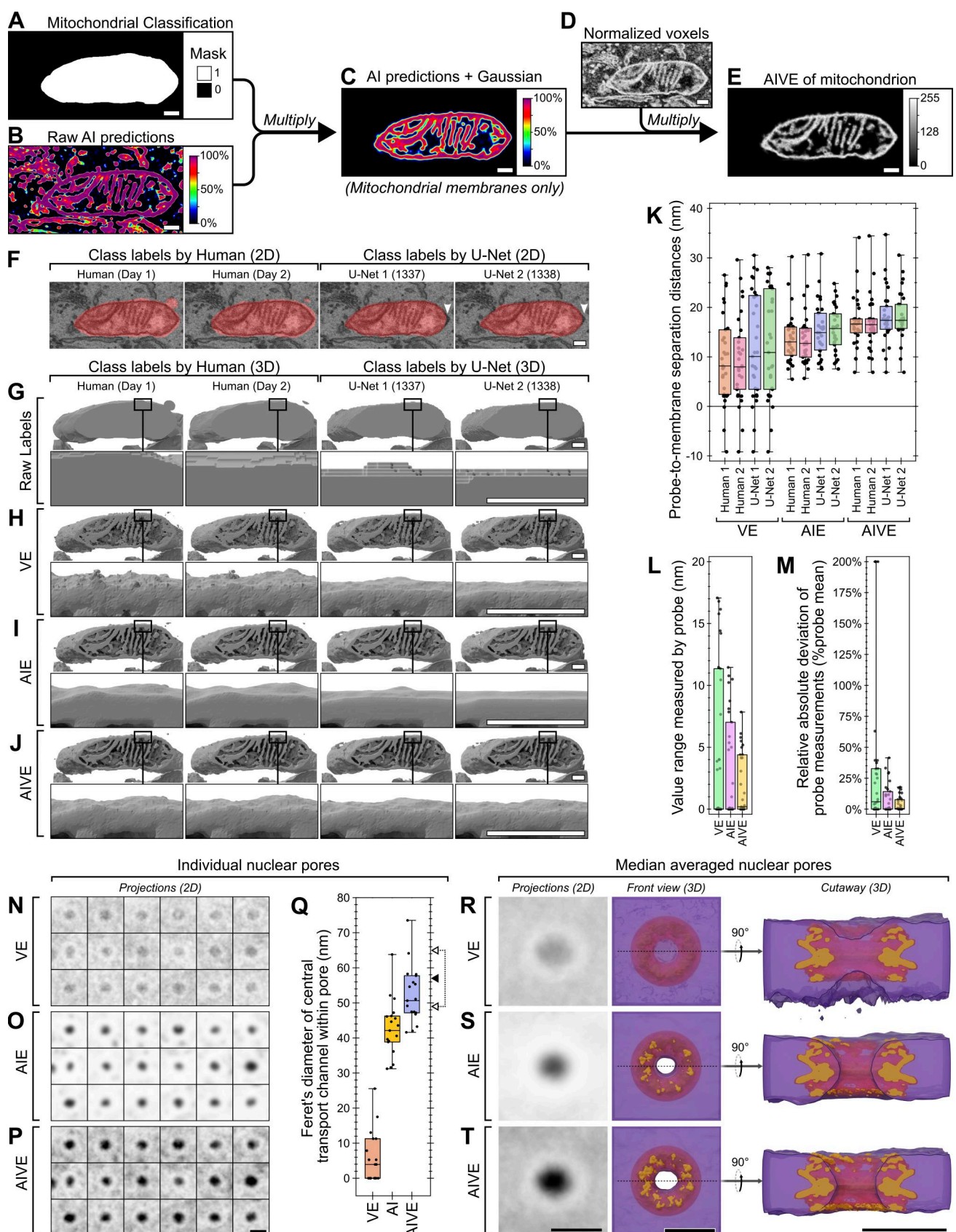

Figure 3. **Object classification does not alter the precision or accuracy of segmentation by AIVE. (A–E)** Flow chart depicting object classification during AIVE, in which (A) organelle labels and (B) membrane predictions are multiplied to generate (C) a contextually masked membrane prediction, (D) which is then combined with the normalized EM data to generate (E) AIVE data belonging to a specific organelle. **(F)** 2D examples of mitochondrial classification labels

provided by one human on different days or two 3D U-Nets with different initialization seeds. **(G–J)** 3D-rendered results classified using the labels shown in Fig. 3 F, after classification via (G) conventional instance segmentation or by using the binary labels (H) for selective VE, (I) AIE, or (J) AIVE. **(K–M)** Quantitative analysis of **(K)** the distance between mitochondrial membranes and each mitochondrial 3D probe (probe locations shown in Fig. S1 D; also see Video 2 for rotation animation depicting probe positions), (L) the range of values recorded by each 3D probe, and (M) the relative absolute deviation in 3D probe measurements for all datasets shown in Fig. 3, H–J. **(N–P)** 2D maximum intensity projections of 18 individual nuclear pores, which were spatially aligned prior to processing and segmentation via (N) VE, (O) AIE, or (P) AIVE (unprocessed pores are shown, and pore locations are shown in Fig. S1, E and F, respectively). **(Q)** Feret's diameter measurements of the central transport channel from each nuclear pore after segmentation via VE, AI, or AIVE (black arrowhead indicates diameter reported by Schuller et al. (2021); hollow arrowhead range indicates diameters reported by Zimmerli et al. (2021). **(R–T)** 2D projections and 3D renderings showing the median averaged data generated via (Q) VE, (R) AI, or (S) AIVE (purple), with spatial alignment to the NPC structure (orange) published in 2021 by Schuller et al. (2021). Plot markers in K–L indicate values for individual 3D probes. Plot markers in T indicate measurements for individual nuclear pores. The interquartile range is indicated by the box, median is indicated by the horizontal lines, and the minimum and maximum are indicated by whiskers. Scale bars; A–J, 200 nm; N–S, 100 nm.

mitochondrial interactions with other cellular organelles in three FIB-SEM datasets from HeLa cells that were processed using AIVE (set #1, Fig. 4 B; set #2, Fig. 4 C; set #3, Fig. 4 D; see Video 3). As mentioned previously, the test dataset used in our benchmarking experiments (Figs. 1, 2, and 3) originated from a small region (Fig. 4 A) within one of these datasets. AIVE data can be subjected to a wide range of analyses (Figs. 4, 5, 6, 7, and 8), the simplest being direct quantification of membrane volume (Fig. 4, F–H). Set #1 (654.0 μm³ total volume of dataset) contained the highest proportion of organellar membranes (84.7 μm³; 12.9% of total volume). Due to the prominence of the cellular nucleus in sets #2 and #3, organellar membranes represented a substantially lower proportion of their total volume (set #2, 858.8 μm³ total, 59.4 μm³ organellar membrane; set #3, 640.9 μm³ total, 39.1 μm³ organellar membrane). The nucleus in set #2 is noteworthy (Fig. 4 C), as it also contained multiple examples of nucleoplasmic reticulum tubes spanning the interior of the nucleus (Drozdz et al., 2017). Excluding the nuclear envelope from subsequent analyses reveals striking similarities in the distribution of membrane between organelles across the datasets (Fig. 4, E–G), with the ER being the most abundant category of organellar membrane, followed closely by mitochondrial membranes. Intracellular vesicles and Golgi apparatus were the next most abundant categories of organellar membrane, but Golgi apparatus was not detected in set #2 (Fig. 4 F); coincidentally, set #2 also contained a lower proportion of early and late endosomal membranes than in the other two datasets. To explore these datasets in greater detail, we next turned our attention to individual mitochondria.

### Quantitative analysis of mitochondrial morphology via AIVE

The three datasets (Fig. 4, B–D) collectively contained 186 individual mitochondria. Morphometric analysis of these mitochondria based on their membranes alone would have limited value, since it would not include the mitochondrial interiors or bulk morphology, which is why our machine learning models were also trained to classify materials other than membrane. For example, the prediction results for "Matter" (class 3) are specifically designed to represent any electron-dense non-membranous materials, such as the mitochondrial matrix (Fig. S1 B). This additional class allowed us to "fill" the mitochondria by multiplying the matter class with the mitochondrial class labels, then adding the result to the AIVE membrane outputs for the corresponding mitochondria. In addition, the filling process enables imputation of the matrix of each mitochondrion.

Morphometric analyses, including measurements of mitochondrial volume, membrane surface area, volume of matrix, shape, and length, were applied to all 186 mitochondria. Based on the morphometric analyses, six outlier mitochondrial morphologies were identified (Fig. 5, A–F). This included the smallest mitochondrion, whose total volume was only 0.035 μm³ (Fig. 5 A). As a point of comparison, the smallest mitochondrion was approximately one-hundredth the volume of the largest mitochondrion (compare Fig. 5 A with Fig. 5 B), demonstrating a wide range of mitochondrial size. Unsurprisingly, the largest mitochondrion also possessed the greatest membrane surface area (25.079 μm²), the largest volume of membrane (1.863 μm³), and the largest volume of matrix (1.605 μm³) of any detected mitochondrion (Video 4). Our analyses of mitochondrial morphologies also included two different metrics of mitochondrial "length" (see Video 5), in which the most elongated and longest mitochondria were identified. The most elongated mitochondrion (Fig. 5 D; 11.89 aspect ratio, 9.36-μm longest-optimal path) was identified by measuring mitochondrial ellipsoid aspect ratios (major axis/minor axis; Fig. 5 J), whereas the longest mitochondrion (Fig. 5 E; 2.91 aspect ratio; 11.229-μm long) was determined by skeletonizing the bulk mitochondrial structure, then measuring the longest-optimal path through the skeleton (Fig. 5 K). Mitochondrial sphericity was also quantified (Fig. 5 L and Video 6, left), leading to the identification of the most spherical mitochondrion (Fig. 5 F; sphericity = 0.860). The wealth of morphological data that was obtained also allowed us to identify the most average mitochondrion (Fig. 5 G, and Video 6, right) that had morphological measurements closest to the average value across every metric quantified. Interestingly, we also noted that mitochondrial membrane volume correlated closely (R > 0.99) with matrix volume in all datasets (Fig. 5 I), demonstrating that internal mitochondrial architecture is tightly controlled despite large variations in morphology.

The morphometric analyses also led to the identification of mitochondrial nanotunnels based on the separation of mitochondrial matrix volumes within a mitochondrion (Fig. 5 C and Video 6, center). The presence of mitochondrial nanotunnels in HeLa cells was unexpected since nanotunnels are typically observed in cellular/tissue environments in which mitochondrial movement is physically constrained (Vincent et al., 2017). To show an example of nanotunnels in a physically constrained tissue and demonstrate AIVE's utility in tissue samples, we conducted FIB-SEM and AIVE of murine skeletal muscle tissue

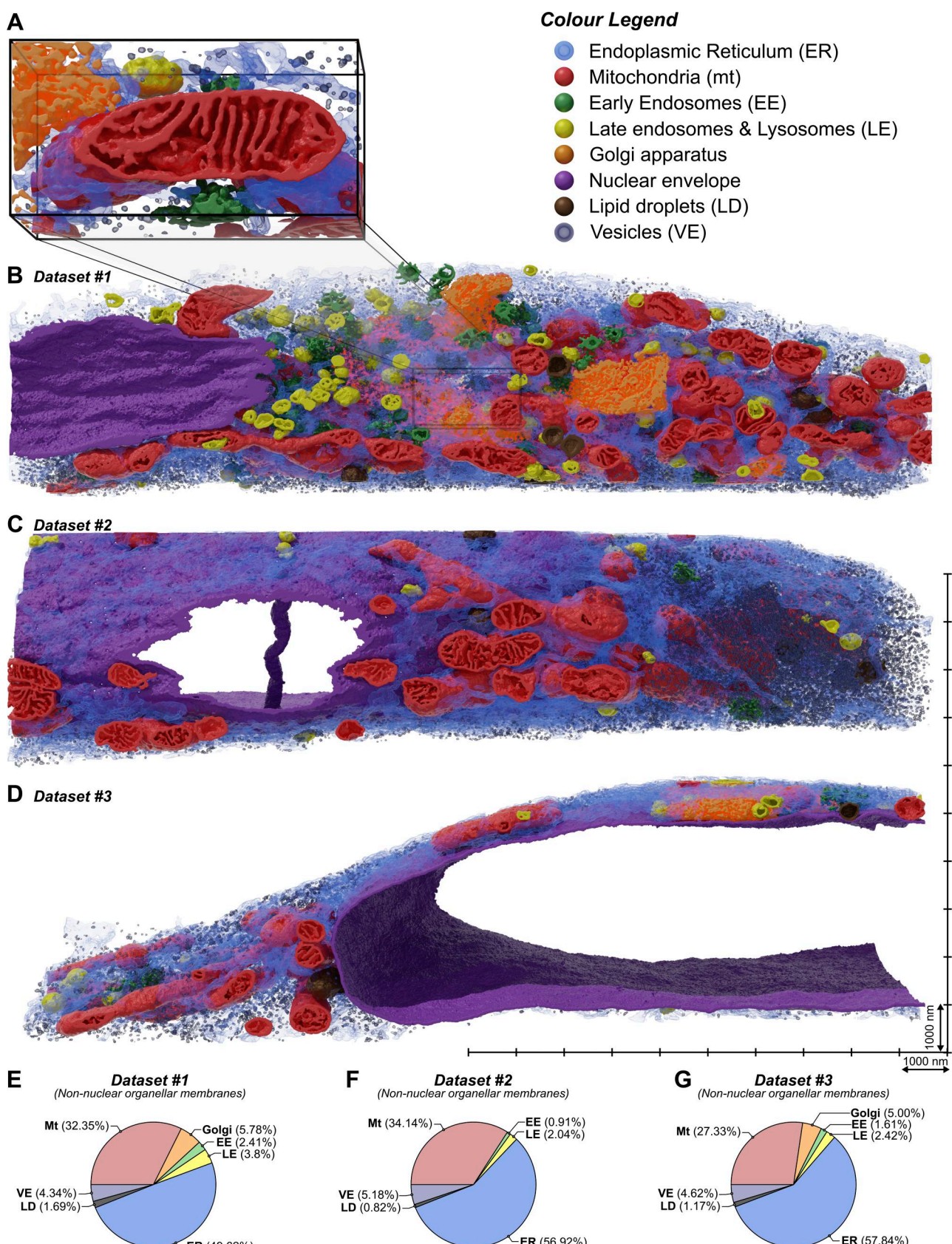

**Figure 4.** **The compositional analysis of membranes in bulk cellular volumes via AIVE. (A–D)** 3D renderings of the three FIB-SEM datasets processed via AIVE for quantitative analysis, showing (A) the test data used in Figs. 1, 2, and 3 alongside overviews of (B) the dataset it originated from (set #1; leading lines indicate location), and **(C and D)** two additional datasets processed via AIVE (set #2 and set #3, respectively). All three overviews are shown at the same scale

(see Video 3 for animated cutaways of the data). **(E–G)** Compositional analysis of membrane volumes belonging to non-nuclear cytoplasmic organelles in (E) set #1, (F) set #2, and (G) set #3. See Video 3 for animated cutaways of overview data. Scale bars; A, the bounding box is 2,400-nm wide; B–D, markers on the scale grid for overviews are 1,000-nm apart.

(Fig. S4). Numerous examples of mitochondrial nanotunnels were observed (Fig. S4, A–C and Video 7), in which mitochondria remained interconnected over distance by reaching around myofibrils via nanotunnels (Fig. S4 and Video 7). In addition, we observed extensive sarcoplasmic reticulum networks that contacted mitochondrial nanotunnels at sites around myofibrils. Collectively, these analyses show the extensive mitochondrial morphology analyses that can be conducted using AIVE across cultured cells and tissue.

### Quantitative analysis of the mitochondrial organelle interactome via AIVE

Next, we aimed to investigate the interactions between mitochondria and other organelles, including MCS, by quantifying the separation distances of each mitochondrion from the ER, early endosomes (EE), late endosomes (LE), lipid droplets (LDs), and general cytosolic vesicles (VE) (Fig. 6). By individually indexing each mitochondrion, we were also able to quantify the separation between each mitochondrion and the remainder of the mitochondrial network.

First, we determined the closest approach made between organelles by measuring the minimum separation distance for each mitochondrion relative to each category of organelle (Fig. 6, A–F). Closest approach values for the EE, LE, and LD organelle categories varied greatly between mitochondria (Fig. 6, C–E), which were often several microns away from these categories of organelle (Fig. 6, C–E). Next, the number of mitochondria in contact with each category of organelle was determined by applying the standard criteria often described for MCSs of <35-nm separation (Jing et al., 2019). On average, we found that 12.13% of mitochondria were in contact with an EE, 24.96% were in contact with a LE, and 10.24% were in contact with an LD (Fig. 6 G). In contrast, every mitochondrion was found to be in contact with the ER (Fig. 6 G), and the mitochondria-ER closest approach distances never exceeded 5 nm (Fig. 6 B). These results are in close agreement with the findings of Friedman et al. (2010), who reported that all cellular mitochondria maintain continuous contact with the ER. Additional categories of mitochondrial membrane contact were also observed, and they included interactions with other mitochondria and cytosolic vesicles (Fig. 6 G). These results demonstrate the utility of AIVE in measuring organelle interactomes while also revealing the extent of organelle contacts and how they vary across different organelles.

### Mitochondrial intrusions are morphological platforms for organelle interaction

Using distance-based criteria to define MCS is a valuable tool. However, we also wanted to account for the magnitude of the contact area relative to the size of an organelle because it gives context to the breadth of a MCS. For example, small point-like interactions with randomly distributed vesicles (Fig. 6 G) cannot

be distinguished from a membrane contact that covers a substantially greater surface area. We therefore sought to refine our quantitative analyses of MCS by accounting for the size of interaction sites between mitochondria with the ER, vesicles (VE), and other mitochondria (Fig. 6, H–M). By grouping distance measurements into 5-nm brackets ranging between 0 and 250 nm, the percentage of a mitochondrion's surface from each separation distance was determined (Fig. 6, K–M). The analysis revealed a prominent peak in the 0–5-nm distance bracket for mitochondrial (Fig. 6 K) and ER distance measurements (Fig. 6 L; Arrowheads), but not for cytosolic vesicles (Fig. 6 M) that were typically further away from mitochondria and occupied little of the mitochondrial surface.

The value peak detected in the 0–5-nm range caught our attention because it is a large surface of contact that cannot occur between round convex spheroids (Fig. S5, A and C), unless one of those objects partially envelops the other (Fig. S5, B and D). Indeed, our analysis of the 0–5nm membrane contacts revealed striking mitochondrial morphologies that enabled membrane contact via organelle intrusion (Fig. 7). Importantly, this characteristic peak between 0 and 5 nm was unlikely to be detected via conventional AI-based binary segmentation alone, since binarized data could not have measured distances between 0 and 2.3 nm (Fig. S1, G–I), and the variation in AI-defined boundaries greatly exceeded 5 nm (Fig. 2 I). The precision conferred by AIVE was therefore essential to the discovery of mitochondrial intrusions as a form of membrane contact.

The intrusions were characterized by an invagination of the outer mitochondrial membrane to form a narrow cavity that protruded into the interior of a mitochondrion (Fig. 7; see Videos 8 and 9). Mitochondrial intrusions frequently contained ER membranes (Fig. 7, A–C and Video 8) that typically occupied the entire intrusion cavity. Interestingly, the ER membranes often displayed higher levels of osmium staining (EM data; Fig. 7, A–C), indicating that the membranes are protein rich or have a unique lipid profile. Mitochondrion-to-mitochondrion intrusions were also observed, in which the exterior membranes of one mitochondrion intruded into an adjacent mitochondrion (Fig. 7, D and E; and Video 9). Unlike the ER-to-mitochondrion intrusions, membranes in this category of intrusion did not display increased osmium staining (Fig. 7, D and E). We also identified a mitochondrion that was intruding into another mitochondrion while simultaneously receiving an intrusion from the ER (Fig. 7 D and Fig. 8 A; see box indicating set #1, mitochondria 61 and 89). This demonstrates that mitochondria can simultaneously give and receive intrusions, potentially enabling inter-organelle communication. Mitochondrial intrusions lacking an intruding membrane within the cavity were also observed (Fig. 7 F), indicating that intrusion cavities may form independently from an intruding membrane.

We asked why mitochondrial intrusions have not previously been reported using established EM approaches. Additional morphological analyses revealed key features that can provide

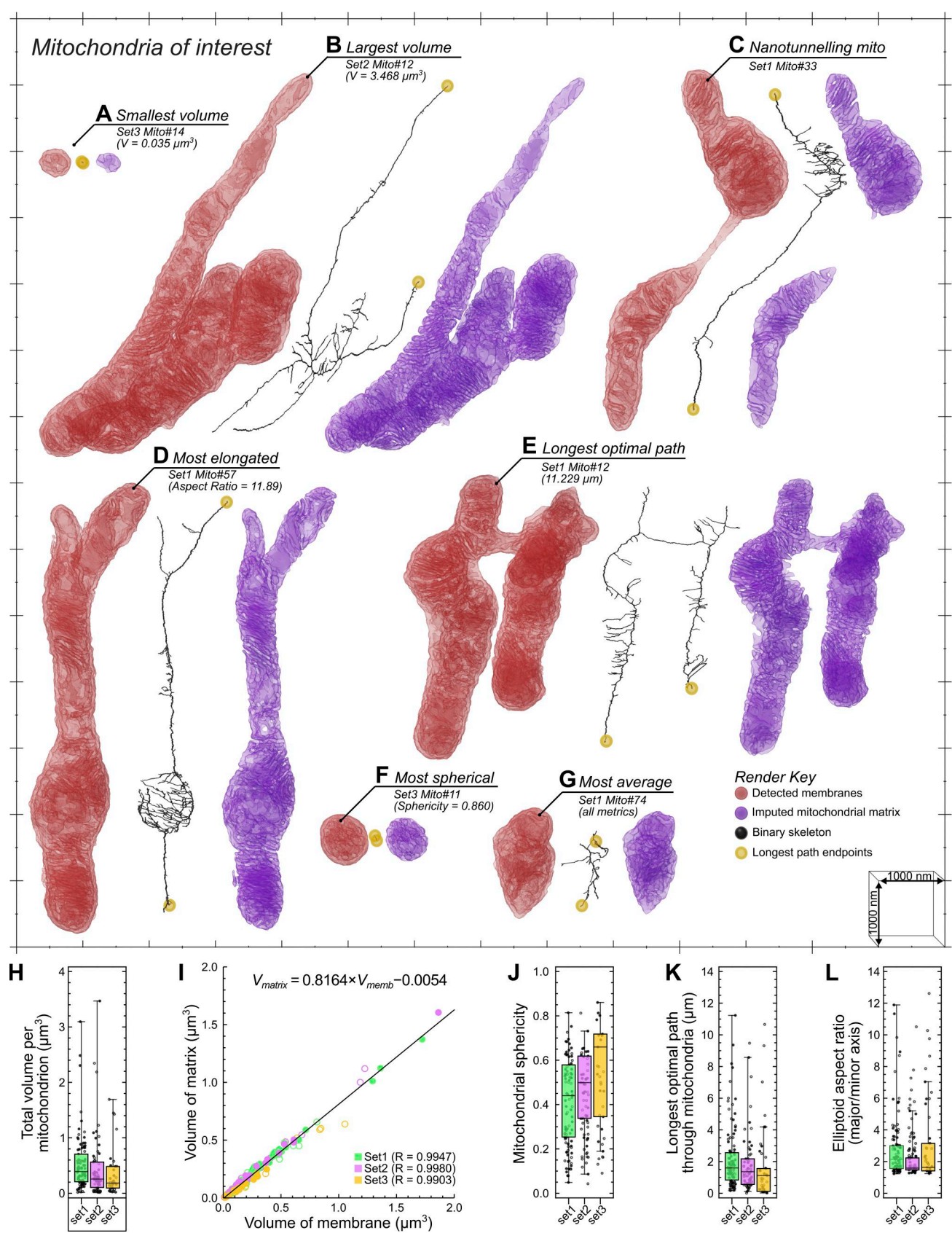

Figure 5. **The morphometric analysis of individual mitochondria via AIVE. (A–G)** Mitochondria of interest that were detected in the AIVE-processed datasets are shown in Fig. 4, showing the membranes of each mitochondrion (red) alongside its matrix (blue) and the binary skeleton calculated for its bulk morphology (black; longest-optimal path endings shown in yellow); these include (A) the smallest mitochondrion, (B) the largest mitochondrion, (C) a

nanotunnelling mitochondrion, (D) the most elongated mitochondrion, (E) the mitochondrion with the longest continuous tubule, (F) the most spherical mitochondrion, and (G) the mitochondrion closest to the average values detected by all morphological metrics. For animations of these data, see Video 4 (for A and B), Video 6 (for C, F, and G), and Video 5 (for D and E). **(H–L)** Quantitative analysis of key morphological metrics of all mitochondria in each set: (H) box whisker chart for the total volume of each mitochondrion, (I) a scatter chart showing the relationship between the volume of membrane (x axis) and volume of matrix (y axis) in each mitochondrion, with (J–L) box whisker charts showing additional metrics for (J) mitochondrial sphericity, (K) mitochondrial length, and (L) mitochondrial elongation. See Video 4. Plot markers indicate value for individual mitochondria. Hollow plot markers indicate mitochondria that were intersected by the dataset boundary; these mitochondria were excluded from analyses sensitive to incomplete mitochondria (Fig. 5, A, E, D, G, and J–L). The interquartile range for mitochondrial population is indicated by the box, the median is indicated by the horizontal lines, and the minimum and maximum are indicated by whiskers. Scale bars; A–G, markers on the scale grid are 1,000-nm apart.

---

an explanation why intrusions have not been observed until now. Of the 186 mitochondria surveyed, we detected a total of 50 mitochondrial intrusions among 36 individual mitochondria (Fig. 8 A), equating to 0.269 intrusions per mitochondrion on average. No mitochondrial morphological traits appeared to correlate with the presence of an intrusion (Fig. 8 B), implying that the formation of a mitochondrial intrusion is stochastic. Indeed, the average number of intrusions per mitochondrion (0.269) was used to model a Poisson distribution that closely approximates the observed values (Fig. 8 C). Compared with the mitochondrion, the internal volume of the average intrusion cavity was minuscule (Fig. 8 D), and the largest sphere capable of fitting within these cavities would have a radius no larger than 50.6 nm (Fig. 8 E), yet they often extended several hundred nanometers inward (Fig. 8 F).

A mitochondrial intrusion can only be confidently identified when viewed in its entirety. This is because without evidence of the entrance, an intrusion would be indistinguishable from a membrane inclusion (see Fig. 7 D), and without evidence of the deeper intrusion, the cavity entrance would appear to be a slight membrane indentation (see Fig. 7 A). We sought to demonstrate this by calculating the probability of detecting a fully intact mitochondrial intrusion via TEM imaging by using Buffon's needle problem (Robertson and Siegel, 2018). Buffon's needle problem is a geometrical probability and is formulated as follows: If a needle of known length (L) is randomly dropped onto an array of regularly spaced lines separated by a known distance (d), what is the probability that the needle will overlap a line after landing (Fig. S5 E)? In the present context, the "needles" represent mitochondrial intrusions with an average length (L) of 367.5 nm (Fig. 8 F), the space between lines represents TEM sections of known thickness (d), and the needles that overlap a line represent incomplete intrusions that extended into adjacent sections. The probability of detecting a fully intact intrusion can therefore be considered the inverse of Buffon's needle problem (Fig. 8 G and Fig. S5 E). For a hypothetical 100-nm TEM section (d = 100 nm) containing a mitochondrial intrusion, there is less than a 1 in 10 chance (P = 0.087) that the intrusion will remain fully intact within that section (solid line in Fig. 8 G). Under these conditions, an observer would be twice as likely to observe only half of the intrusion (P = 0.177964; see dashed line, Fig. 8 G), which would be indistinguishable from a membrane inclusion or indentation of the outer membrane. Even if the section thickness was equal to the length of an intrusion (L = d), the probability of the structure being intact remains low (P = 0.363; solid line in Fig. 8 G). It is important to note that the real probability of observing an intrusion is likely even lower, since they were only

present in a fraction of the mitochondria (Fig. 8 C). In contrast, there would be a 95% chance (P = 0.953) of an intrusion remaining intact if the Z-depth of the sampled volume spanned 5 μm (d = 5,000 nm), which would not be possible using TEM but easily achieved via FIB-SEM. These calculations demonstrate why mitochondrial intrusions have not previously been identified using historical TEM.

Taken together, by using AIVE, we were enabled to discover that HeLa cell mitochondria can contact the ER (and other mitochondria) via intrusion sites (Figs. 7 and 8), which provide a larger interface for surface interactions not possible between round objects (Figs. 7 and S5). We have also shown that these intrusion sites are more likely to be indistinguishable from a mitochondrial membrane inclusion when viewed by TEM (Fig. 8 G).

## Discussion

In this study, we have outlined and benchmarked a technique termed AIVE for volumetric EM. AIVE enables electron signals detected via FIB-SEM to be processed for rapid 3D reconstruction and analysis with high fidelity. AIVE's fidelity is achieved by using the ground truth of EM data in the final output. In AI research, ground truth is used to describe the preferred result from a trained model (Lebovitz et al., 2021), but the original definitions of ground truth specify that it originates from objective empirical measurements (Woodhouse, 2021). In the context of this manuscript, the backscattered electron signals acquired via FIB-SEM imaging represented the most objective and empirical measurements that could be used for ground truth. AIVE therefore uses an objective definition of ground truth by linking AI predictions to real empirical measurements. Since its inception, AIVE has been applied to volumetric EM analyses of cultured cells (Lee et al., 2024; Nguyen et al., 2021) and animal tissues alike (Fig. S6). However, in the absence of benchmarking, it was unclear why a cell biologist might choose to apply AIVE versus AI predictions alone. Here, we demonstrate that membrane boundaries are the least certain areas of AI predictions. Through extensive benchmarking, we make a case for the benefits of AIVE to define membrane boundaries with high fidelity that provides benefits for the study of organelle ultrastructure and MCSs. Moving forward, it would be beneficial to integrate the process of AIVE as a default option in AI-assisted segmentation strategies. As it stands, AIVE is a separate step to enhance AI-assisted predictions, but through incorporation within segmentation algorithms AIVE can be seamlessly applied in the future.

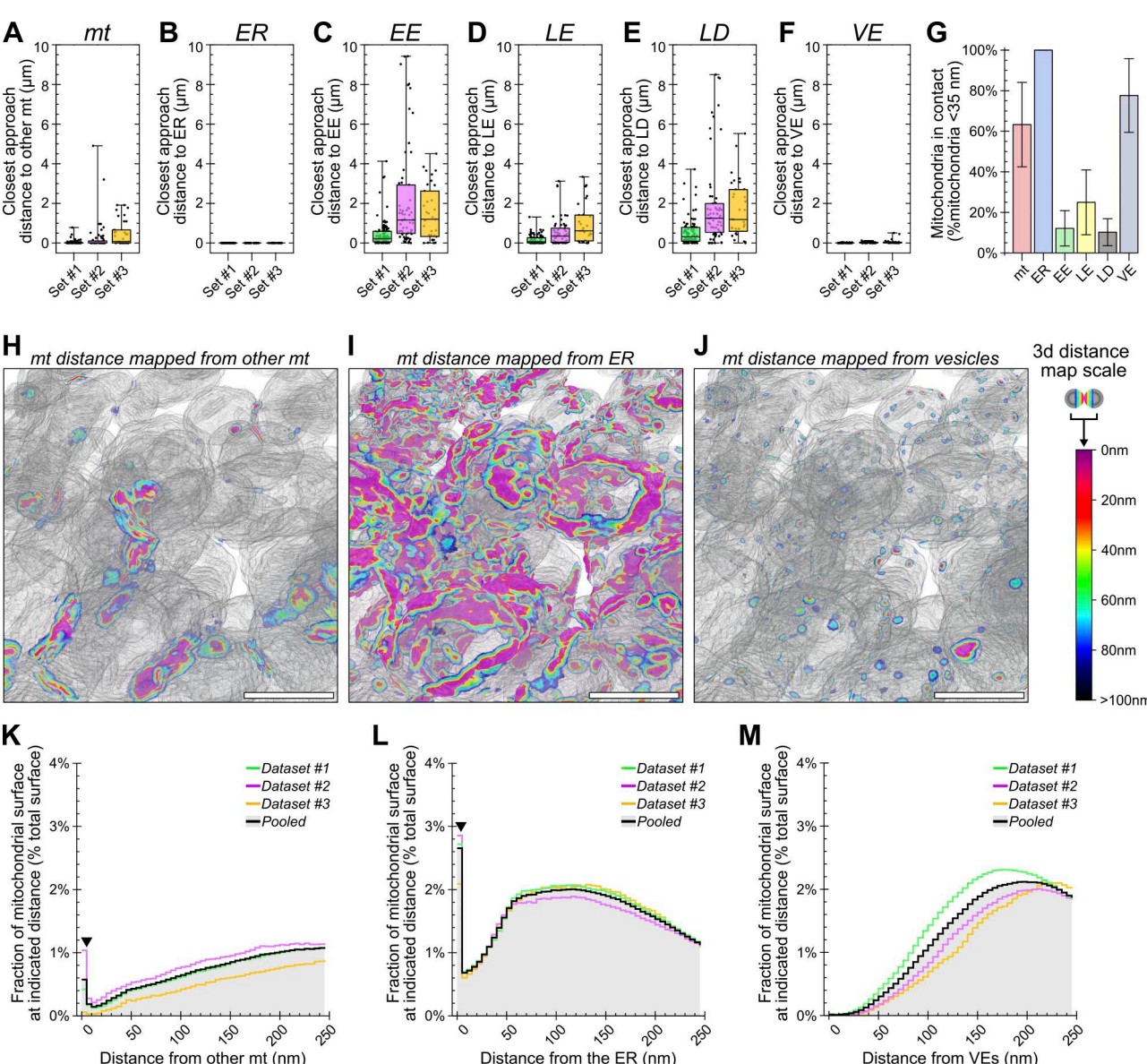

Figure 6. **AIVE for the analysis of the mitochondrial interactome. (A–F)** Quantification of the closest approach distance separating each individual mitochondrion from (A) the rest of the mitochondrial network, (B) the ER, (C) EEs, (D) LEs, (E) LDs, or (F) cytosolic vesicles. **(G)** Average percentage of all mitochondria detected within 35 nm of the indicated organelle per set. **(H–J)** 3D rendering of distance-mapped mitochondrial surfaces from a subregion in set #1. The mitochondrial surfaces are shaded with a colorimetric scale indicating surface distances from (H) other mitochondria, (I) the ER, or (J) cytosolic vesicles. **(K–M)** Surface area histograms for the mitochondria located within 100 nm of the indicated organelle, depicting the average percentage of a mitochondrial surface located within indicated distance of (K) other mitochondria, (L) the ER, or (M) cytosolic vesicles (arrowheads in K and L indicate the anomalous peak, explained in Fig. S5, A–D). Plot markers in A–F indicate value for individual mitochondria. The interquartile range is indicated by the box, the median is indicated by horizontal lines, and the minimum and maximum are indicated by whiskers. Data in G are mean ± SD calculated for the averages in each dataset. Histogram data in K–L are mean surface area percentages binned into 5 nm-distance brackets relative to the target organelle; mean values for each set are also shown. Scale bars; H–J, 200 nm. mt, mitochondria.

Mitochondrial membrane ultrastructure is tightly linked to the biochemical functions of a mitochondrion (Zick et al., 2009). For example, mitochondrial damage and dysfunction are linked to a range of abnormal mitochondrial morphologies, including cristae disturbances and network fragmentation (Jenkins et al., 2024; Linda et al., 2009; Vincent et al., 2016). In mitochondrial diseases, abnormal morphologies can manifest as osmiophilic inclusions of concentric membranes and "onion-shaped" cristae within the mitochondria (Linda

et al., 2009; Vincent et al., 2016), while in other cases, cristae structures are diminished (Siegmund et al., 2018). Mitochondrial ultrastructure can also change in response to the metabolic activity of mitochondria (Ryu et al., 2024), in which mitochondria undertaking oxidative reactions retain defined cristae, whereas those enriched in reductive reactions lack cristae. The intimate association between mitochondrial structure and function lends itself to exploration using volumetric EM. AIVE can therefore be used in various contexts,

Figure 7. **Mitochondrial intrusion sites facilitate membrane contact with the ER and other mitochondria. (A–C)** Examples of ER-to-mitochondrion intrusion sites detected in (A) set #1, (B) set #2, and (C) set #3, displayed as partial (50 nm) 2D-averaged projections of raw EM and AIVE data (left panels), rendered in 3D (center panels), and with a colorimetric distance map relative to the ER (right panels). For animations of these data, see Video 8. Additional examples are provided in Fig. S6, A–C. **(D and E)** Examples of mitochondrion-to-mitochondrion intrusion sites detected in (D) set #1 and (E) set #2, displayed as partial (50 nm) 2D-averaged projections of raw EM and AIVE data (left panels), rendered in 3D (center panels), and with a colorimetric distance map relative to

other mitochondria (right panels). For animations of these data, see Video 9. Additional examples are provided in Fig. S6, D and E. **(F)** 3D-rendered examples of empty intrusion sites detected in each dataset. Scale bars, 200 nm; bounding dimensions of each 3D dataset frame, 1.5 × 1.5 × 1.5 µm.

ranging from understanding mitochondrial ultrastructure in different tissues and disease states to understanding the relationship between mitochondrial morphology and physiological and biochemical status. Cristae are intricate structures of the mitochondrial inner membrane that come in all manner of unusual shapes and sizes depending on cell type and metabolic status (Siegmund et al., 2018). For example, cristae in astrocytes appear to take the shape of triangles, whereas cristae in adrenal cortex cells are highly circular (Zick et al., 2009). Cristae analyses such as these have typically been conducted using 2D TEM. By applying vEM and AIVE, we anticipate that additional fascinating features of cristae structure and biology will be revealed.

Mitochondrial structures can dynamically adapt according to their cytosolic environment. For example, mitochondria in muscle tissue form nanotunnels that enable connectivity of the mitochondrial network around the physical obstruction of myofibrils (Huang et al., 2013; Vincent et al., 2017) (Fig. S6). Nanotunnel connectivity can increase in murine cardiac myocytes with defective calcium release from the sarcoplasmic reticulum (Lavorato et al., 2017), indicating an adaptive role for nanotunnels. Through AIVE, we were able to expand on previous studies of nanotunnels and show the extensive ultrastructure of mitochondrial nanotunnel networks in murine skeletal muscle (Fig. S6 B). We also revealed extensive MCSs between nanotunnels and the sarcoplasmic reticulum (Fig. S6 C) that may play a role in calcium buffering during contraction (Lavorato et al., 2017). Our findings demonstrate the benefits of collecting large EM volumes combined with AIVE in identifying long-distance organelle connections, including mitochondria via nanotunnels and the nucleus via nucleoplasmic tubes (Fig. 4 C).

Another area of cellular biology illuminated by AIVE and vEM was the visualization and quantitation of MCSs. Through AIVE, we were enabled to undertake high-resolution analyses of organelle interactomes via MCSs (Figs. 5, 6, and 7). The value of AIVE for MCS analyses was best demonstrated by the discovery of a hitherto unknown form of mitochondrial contact that we term the mitochondrial intrusion (Fig. 7). Through intrusions, mitochondria were observed to contact other mitochondria or the ER. Intrusion relays in which a mitochondrion simultaneously contacted the ER and other mitochondria were also observed. These interactions go beyond the binary MCSs that are typically described and highlight the complexity of organelle contacts that can be revealed by vEM and AIVE. Indeed, in a previous application of AIVE with vEM, wholesale organelle-ome changes were observed in response to perturbation of a single organelle family (Lee et al., 2024). We note that mitochondrial intrusions form basally, and they are distinct from previously reported mitochondrial cavities (Miyazono et al., 2018). Mitochondrial cavities, induced by fragmenting mitochondria through depolarization, differ greatly from intrusions in size and morphology and in contact characteristics, including the lack of intramitochondrial contacts (Miyazono et al., 2018).

What might be the benefits of mitochondrial intrusions as a form of MCS? One answer is increased surface area for interaction. The exterior of a mitochondrion is typically round and convex (Giacomello et al., 2020; Lackner, 2019), but this is the least efficient geometry for mediating surface-to-surface interactions (Hertz, 1882) (Fig. S4 A). Mitochondrial intrusions can maximize the surface area of an MCS (Fig. S4 B) and therefore promote the efficiency of inter-organelle communication (Scorrano et al., 2019). Another potentially important benefit of intrusions is that the increased contact area can serve as an anchor to stabilize an otherwise fleeting MCS. Once formed, the very close proximity of membranes within intrusions (0–5 nm) can feasibly support tethers that support lipid transfer, including the mitochondrial proteins PTPIP51 and MIGA2 (Freyre et al., 2019; Kim et al., 2022; Yeo et al., 2021), which could be facilitated by Mfn1/2 for intramitochondrial intrusions and Mfn2 and VAPB for ER-mitochondria intrusions (De Vos et al., 2012; Naon et al., 2023; Stoica et al., 2014). In contrast, MCSs involving inositol 1,4,5-triphosphate receptors that facilitate calcium transfer between ER and mitochondria might be too bulky for the very close contacts made by intrusions (Csordas et al., 2010; Voeltz et al., 2024).

Overall, we demonstrate the utility of AIVE combined with volumetric EM for understanding the ultrastructure and interactome of cellular organelles, while also demonstrating its capacity to reveal fascinating and unexpected features of cell biology, including the identification of mitochondrial intrusions as a form of MCS.

## Materials and methods
### Cell culture
HeLa cells (RRID:CVCL_0030) were cultured in DMEM supplemented with 10% (vol/vol) FBS (Cell Sera Australia), 1% penicillin-streptomycin, 25 mM HEPES, 1x GlutaMAX (Life Technologies), and 1x nonessential amino acids (Life Technologies). Samples were prepared by culturing HeLa cell monolayers on a polymer film substrate (Aclar film; 203.2-µm thick; ProSciTech), which was heat-welded to the plastic surface of a 10-cm tissue culture dish with a soldering iron. Three dishes were prepared and UV sterilized before seeding ~6 × 10⁶ HeLa cells onto the Aclar film in each dish. The cells were allowed to attach under normal culture conditions for 48 h, with replacement of the culture medium 1 h prior to fixation.

### Sample preparation of HeLa cells for FIB-SEM
The samples were chemically fixed with 4% PFA in 0.1 M phosphate buffer (pH 7.2) at 37°C for 1 h, with overnight postfixation with 2.5% glutaraldehyde in 0.1 M sodium cacodylate buffer at 4°C. Each polymer film was detached from their tissue culture dish before being transferred to a polypropylene tray (lid of a pipette tip box), where they were immobilized by heat-welding with a soldering iron. A BioWave Pro microwave system (Pelco)

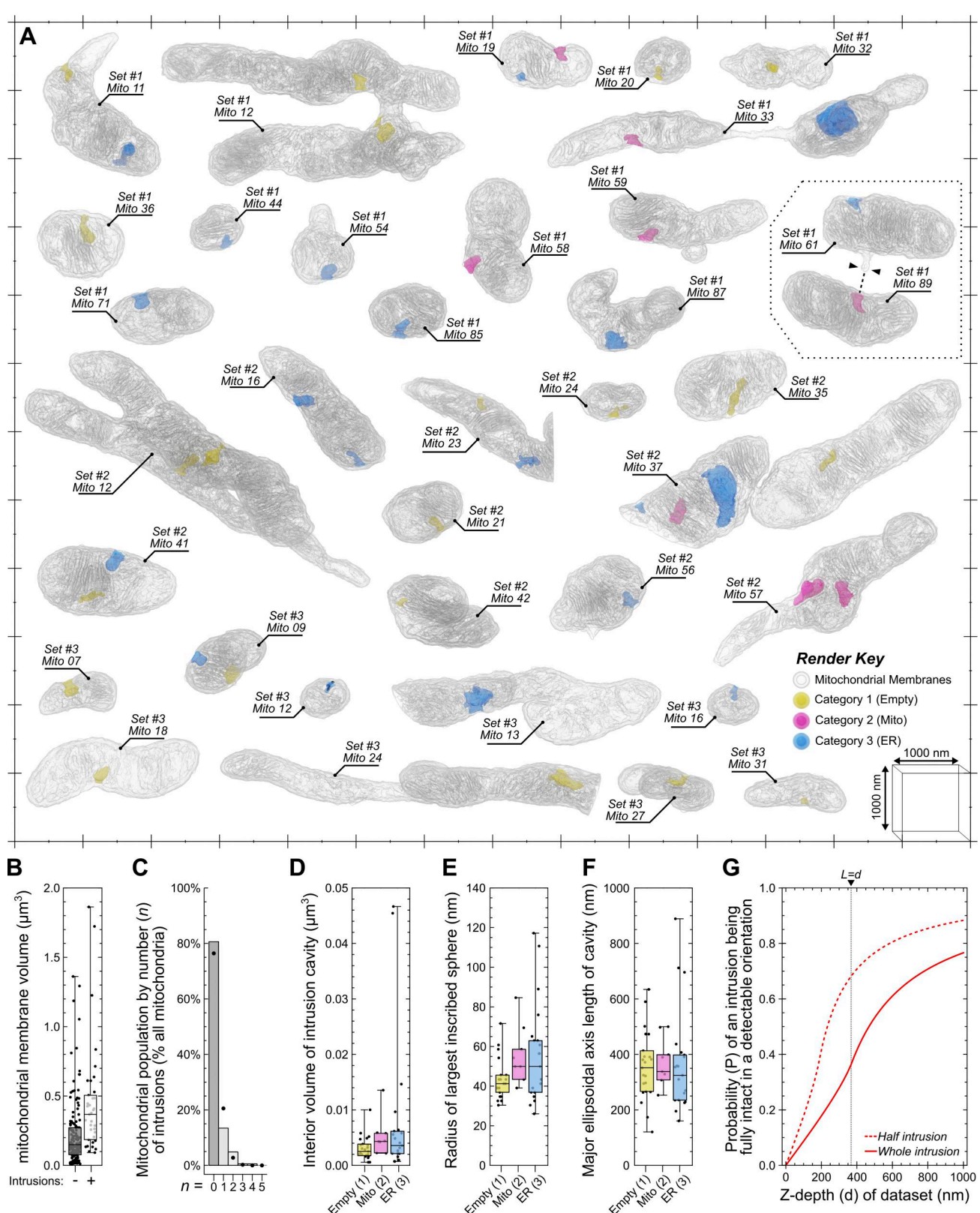

Figure 8.   **vEM is essential for the analysis of mitochondrial intrusion sites. (A)** 3D-rendered examples of mitochondria containing intrusion sites, with their interior cavities colorized by the category of intrusion. **(B)** Quantitative analysis comparing the membrane volume of mitochondria in the presence or absence of mitochondrial intrusions. **(C)** Bar chart showing percentage of the mitochondrial population by their number of intrusions, with plot markers indicating expected values calculated by a Poisson distribution of the same average. **(D and F)** Quantitative analyses of each intrusion cavity by category, showing their (D) total interior volume, (E) radius of the largest inscribed sphere they could enclose, and (F) the length of their major ellipsoidal axis. **(G)** Statistical modeling of Buffon's needle problem, showing the probability of detecting an intact mitochondrial intrusion (solid line) or only half an intrusion (dashed line)

when an intrusion is present, as a function of the Z-depth ("d") in a hypothetical FIB-SEM dataset or TEM section. Plot markers in B indicate value for individual mitochondria, and in D–F indicate value for individual intrusion cavities. The interquartile range for each chart is indicated by the box, the median is indicated by the horizontal lines, and the minimum and maximum are indicated by the whiskers. Scale bars: Major markers on the scale grid surrounding mitochondria are 1,000-nm apart, and minor markers are 500-nm apart.

was used for microwave-assisted sample processing in all subsequent stages. Osmication was conducted using a modified OTO method (Seligman et al., 1966) with three microwave duty cycles (120-s on, 120-s off) at 100 W under vacuum where indicated in the following stages: 2% (wt/vol) $OsO_4$, 1.5% (wt/vol) $K_3Fe(CN)_6$ in 0.1 M cacodylate buffer (pH 7.4) at 4° for 2 h, followed by microwave processing; three MilliQ water rinses; 1% (wt/vol) thiocarbohydrazide in water with microwave processing; three MilliQ water rinses; and then 2% $OsO_4$ in water with microwave processing. After further MilliQ water rinses, the samples were en bloc stained with 2% (wt/vol) aqueous uranyl acetate (Silva et al., 1968), followed by Walton's lead aspartate (Walton, 1979), with each stain requiring three microwave duty cycles (120-s on, 120-s off) at 100 W under vacuum. Microwave assisted dehydration (150 W for 40 s per stage) was performed by graduated ethanol series (80%, 90%, 95%, 100%, and 100% [wt/vol]) followed by propylene oxide (100%, 100% [wt/vol]). The samples were infiltrated with a modified Araldite 502/Embed 812 resin of the following composition (described for 10 ml resin): 2.20 g Araldite 502, 3.10 g Embed 812, 3.05 g nadic methyl anhydride, 3.05 g dodecenylsuccinic anhydride, and 400 μl benzyldimethylamine. Infiltration was conducted using a graduated concentration series in propylene oxide (25%, 50%, 75%, 100%, and 100% [vol/vol]; 180 s at 250 W under vacuum). Final embedding was conducted with minimal resin, which was polymerized at 60°C for 48 h.

The Aclar film was peeled from each of the resin-embedded cell monolayers before using a hammer to subdivide the resin into a random assortment of shards that were pooled into a single collection. Sample shards were blindly taken from the sample pool, deposited on a small droplet of the Araldite 502/Embed 812 resin on the tip of a 3.2-mm diameter aluminum rod; another droplet of resin was then placed on top of that shard so that another randomly selected shard could be placed on top, and the process was repeated to embed sandwiches of randomly oriented cells at least three monolayers thick before final polymerization at 60°C for 24 h. The stacked cell monolayers were hand-trimmed with a razor blade before trimming with a glass knife to expose multiple monolayers for 3D FIB-SEM imaging.

### Data acquisition and preprocessing
FIB-SEM imaging was conducted three separate times on randomly selected cells; one cell was chosen from each cell monolayer. Imaging was conducted using a cryo-Helios G4 UX FIB-SEM (FEI) at room temperature using the TLD detector in immersion lensing mode for backscatter electron imaging at 3.255-nm per pixel (2 kV, 100 pA, 3-μs dwell time). Ion milling was conducted at 10-nm per slice (gallium, 30 kV, 9 nA) using the Auto Slice and View (v4.1; FEI) software. All datasets were spatially registered (rigid transform) using the virtual stack registration plugin in FIJI (v1.53 t; RRID:SCR_002285) (Arganda-Carreras et al., 2006; Rueden et al., 2017). The signal-normalized inputs for AIVE were generated using a pseudo-3D

implementation of CLAHE, in which three different 2D CLAHE calculations were made from the front (XY), top (XZ), and side (YZ) of the registered dataset. Each 2D calculation was made using a modified version of the CLAHE plugin released by Stephan Saalfeld in 2009; this modified plugin accounts for the anisotropy of voxels in the XZ and YZ directions by allowing the use of a rectangular sliding window during CLAHE. This is distinct from the earlier variations of AIVE (Nguyen et al., 2021), which did not account for voxel anisotropy during the CLAHE calculation. The modified plugin ("CLAHE_Anisotropic.class") and script for batch automation ("CLAHE-Batch-3DCLAHE-AnisotropicXYZ.ijm") are both provided on GitHub (https://github.com/BenPadman/AIVE.git).

### Sample preparation and AIVE of mouse muscle tissue
Wild-type C57BL/6J mice (RRID:IMSR_JAX:000664) were anesthetized at 3 wk of age prior to dissection of the tibialis anterior muscle, which was immersion fixed in 4% PFA in 0.1 M phosphate buffer at room temperature (1 h). Tissue was further dissected under fixative to extract 1-mm cubed samples, which were immersed in secondary fixative (4% PFA and 2.5% GA in 0.1 M phosphate buffer) for 24 h. All subsequent sample processing stages and imaging conditions were conducted as described above for cultured cell samples. Two separate FIB-SEM datasets of skeletal muscle were acquired. A machine learning model was trained on three classes of voxels designated "Cytosol," "Fibrils," and "Membrane," using 120,000 samples per class (360,000 total) to train the final model with a 10-fold test cross-validation result of over 97% self-accuracy. Mitochondria were manually classified in Microscopy Image Browser (MIB) as described above, and all remaining membranes were designated as sarcoplasmic reticulum. The final AIVE results were calculated as described above and reconstructed in 3D by marching cubes with an isovalue of 64. All animal work was approved by the Monash Animal Research Platform Ethics Committee (#17628; MARP), Monash University, Melbourne, Australia, and conducted in compliance with the specified ethics regulations.

### Computational hardware
All presented analyses were conducted on one computer equipped with consumer-grade hardware, which included an Intel Core i9-9900KF CPU clocked at 3.60 GHz (16 cores), 64 Gb of RAM at 2133 MHz, one 2-Tb SSD (Samsung 970 EVO Plus) for short-term storage during machine learning feature calculation, two 8-Tb HDD's in RAID1 for long-term storage of machine learning features and results, and an Nvidia GeForce RTX 3080 Ti for deep learning and 3D rendering.

### Machine learning
Machine learning was conducted within the Waikato Environment for Knowledge Analysis (WEKA v3.9; RRID:SCR_001214) (Frank et al., 2004). Training features were extracted from the

registered image data using the 3D Trainable Weka Segmentation (TWS) plugin for ImageJ (RRID:SCR_003070 & SCR_002285) (Arganda-Carreras et al., 2017; Schindelin et al., 2012). As proposed in the original manuscript (Arganda-Carreras et al., 2017), we wrote a series of custom scripts for the TWS plugin to expand and automate its capabilities; these scripts have been provided on GitHub (https://github.com/BenPadman/AIVE.git). The scripts were designed to minimize RAM storage requirements for general machine learning by ensuring that all image features required for training and evaluation are only calculated once prior to local storage on a hard disk. Instead of calculating all features for the entire stack simultaneously, the scripts only inspect a portion of the stack at a time while including the minimum number of flanking slices required by the radius of a 3D image filter. This enables features to be calculated for image stacks of arbitrary length at the cost of wasted computation on flanking slices. Furthermore, the script generates features incrementally by grouping image filter calculations into subsets, temporarily storing the results on a local hard disk, and then merging them to generate final feature stacks for each slice (see script: "ML-Features-PART1-3DFeatSplitter-Sigma8.bsh"). Our approach also includes the calculation of additional custom 2D features (see script: "ML-Features-PART2-2DFeatSplitter.bsh"), which are merged with the original 3D feature stacks (see script: "ML-Features-PART3-Combine3-Dand2DFeatures.bsh"), allowing us to calculate 175 features per slice. The scripts required to execute this process are provided. In a departure from the TWS plugin, all training annotations were generated externally by using the MIB (RRID:SCR_016560) to generate label images where each training class is defined by the numerical value of a voxel (value of 1 for class 1, value of 2 for class 2, etc). A script was then used to import each training label image into TWS with its corresponding feature stack (Fig. 1 G) to extract feature measurements from a predetermined number of randomly selected voxels per class (see script: "ML-CorePartA-ExtractTrainingDataFromFeatures.bsh"). The feature data extracted via this script can then be pooled using the "append" function in WEKA, then imported into WEKA Explore for analysis and machine learning to generate various trained models, as was shown in Figs. 1, 2, and 3. These trained models are loaded by a final script ("ML-CorePartB-ApplyClassifierToFeatures.bsh"), which is designed to evaluate preexisting feature stacks. A complete protocol describing the use of each script is available at Protocols.io (https://doi.org/10.17504/protocols.io.14egn48x6v5d/v1).

These scripts were used for machine learning throughout the manuscript as follows. The analyses in Figs. 1, 2, and 3 used models trained to detect three classes of material: "Sol" (class 1), representing granular protein content of the cytoplasm and nucleus; Matter (class 2), representing electron-dense homogenous materials that do not belong to a membrane (i.e., mitochondrial matrix); and "Memb" (class 3), representing genuine cellular membranes. The division of non-membranous materials into two classes was aimed to account for the class imbalance problem; training annotations were drawn by hand in MIB. For the analyses shown in Figs. 1 and 2, feature stacks for each of the three altered test datasets were generated separately before using a shared pool of training labels to extract feature measurements for 1,500 voxels per class per slice from eight slices (12,000 samples per class, 36,000 samples per stack, 108,000 feature measurements total). For the training of the six models in Figs. 1 and 2: RF, J48 (the Java-compatible extension of Ross Quinlan's C4.5 classifier), multilayer perceptron, decision table, JRip (Java-compatible propositional rule-based RIPPER), and projective adaptive resonance theory neural network. Differences between model outputs were calculated using ImageJ by calculating the absolute difference in predicted values ($\Delta V$) at each voxel between each model and every other model; results were visualized by averaging all $\Delta V$ comparisons for a given model when compared with all others. For 2D analyses (Fig. S3), models using the same architectures as shown in Figs. 1 and 2 were trained on 16,000 samples per class (64,000) using only 2D features.

Data shown in Figs. 4, 5, 6, 7, and 8 used pooled feature data from all three datasets to train a random forest classifier as described above, except that training involved five unique classes of material. The first three of these classes were used to detect non-membrane materials: "Void" (class 1), representing the homogenous extracellular regions that lack an electron signal; Sol (class 2), representing granular protein content of the cell and nucleus; and Matter (class 3), representing electron-dense homogenous materials that do not belong to a membrane (i.e., mitochondrial matrix). The remaining two classes (4 and 5) were subcategories of membrane: Memb (class 4), representing the majority of cellular membranes, and "Vesc" (class 5), representing vesicular membranes; the membrane subclasses needed to be merged together to generate a final AIVE result, but separation into two classes allowed the classification of cellular vesicles by machine learning alone. Protocol describing the above procedures is available at Protocols.io (https://doi.org/10.17504/protocols.io.14egn48x6v5d/v1).

**Organelle classification for AIVE**
Some of the raw classifier outputs described above were exploited to assist the classification of organelles. For example, dilation of the Void class was used to assist classification of the plasma membrane, whereas the Sol class labels in the nucleus were used to assist annotation of the nuclear envelope. Manual labeling in MIB was then used to classify the Golgi apparatus, EEs, LEs, and LDs. The mitochondria were also classified in this way, but multiple subclasses were used to ensure that adjacent unfused mitochondria were considered independently from one another, thus allowing each individual mitochondrion to be indexed for independent analyses. When the U-NET 1 (1,337) from Fig. 3 became available, it was used to selectively refine perimeter annotations for some mitochondria, with all the final annotation decisions being left to the human analyst. The ER was classified through a process of elimination; it was the only remaining unclassified membrane after classification of all other organelles. Final class labels were exported from MIB as a label image. The label images containing all class labels were separated into individual binary stacks, with each image representing one class of organelle. These binary stacks were blurred with a 10-nm Gaussian filter. The 32-bit predictions generated for Memb and Vesc (classes 4 and 5; described above) were combined, then multiplied with the binary stacks and blurred with a

10-nm Gaussian filter before merging the results with 3D CLAHE results calculated earlier. This differs from earlier variations of AIVE (Lee et al., 2024; Nguyen et al., 2021), which treated membranes as a single continuous class of structure. This generates the final AIVE results for each individual organelle class. Protocol describing the above procedures is available at Protocols.io (https://doi.org/10.17504/protocols.io.14egn48x6v5d/v1).

## U-Net training and deep learning

Two U-Nets with 3D anisotropic architecture (same padding) were trained via stochastic gradient descent with momentum ( 0.001 weight decay; 0.9 momentum), using 80 × 80 × 80 voxel input patches (1,040 × 1,040 × 1,600 nm in XYZ), 5 × 5 × 5 filters, leaky ReLu activation layers (0.001 scale), and a dice pixel classification layer. Both U-Nets were structured with 16 input filters and 4 layers, and data augmentations (both 2D and 3D) were applied to 80% of the input patches during training with a piecewise learning schedule (initial learn rate of 0.01; dropping by 80% every 5 epochs) for 35 epochs (512 iterations per epoch). The only difference between these two U-Nets was the random seed value used during network initialization; the first U-Net used a random seed of 1,337, while the second used a value of 1,338 (Fig. 3 F). To minimize segmentation discontinuities between adjacent prediction tiles (Huang et al., 2018), each U-Net was applied using 5% tile overlap with four different orientations of the dataset (original, X-flipped, Y-flipped, and Z-flipped). The results were then returned to their original orientations before averaging to create an ensemble result. Image classification by a human image analyst in the same figure was conducted on two consecutive days and required ~10 min to complete.

## 3D distance probes

To simulate the effect of result variation on MCS measurements, all static distance probes were assigned in accordance with standard criteria for designation of a contact site (<35-nm separation; [Jing et al., 2019]). Initial coordinates for the 3D distance probes were chosen by running a stochastic simulation of diffusion-limited aggregation in Mathematica (version 9; RRID: SCR_014448). These coordinates were annotated in a blank TIFF stack with identical dimensions to the test dataset by converting the voxel located at each indicated coordinate from black (value of 0) to white (value of 255). Probe coordinates further than 35 nm from (or inside) a membrane were removed from the collection by binarizing the membranes shown in Fig. 1 C, dilating them by 35 nm, and then subtracting the un-dilated membranes from the result; any 3D probes outside the boundaries of this result were deleted. This process was repeated until 30 unique probe locations were identified (Fig. 2 H).

Analyses of mitochondrial membranes (Fig. 3, G–M) required an additional set of criteria for the distance probes, since the original 3D probes were generated without relevance to any specific organelle (Fig. 1 H), and most were located further than 35 nm from a mitochondrion. 3D probe coordinates for mitochondrial membranes were therefore generated by using the same 35-nm membrane boundary mask described for general membranes above in combination with an additional mask

representing the mitochondria, as follows: All mitochondrial class labels shown in Fig. 3 F were averaged, binarized, dilated by 35 nm, then applied in a Boolean AND operation with the 35-nm membrane mask used to define the first probe positions (Fig. 1 H). The resulting mask was then used to assign new random points as described above, until 30 new random points were defined within 35 nm of a membrane and within 35 nm of the mitochondrial class label (Fig. 4 K). Value ranges for distance measurements (Fig. 2 I and Fig. 3 L) were calculated by subtracting the minimum recorded distance from the maximum recorded distance for one probe. Relative absolute deviations (also known as relative mean absolute difference) in measurements from each probe (Fig. 2 J and Fig. 3 M) were calculated by dividing the mean absolute difference (difference between each measurement and the mean value) of measurements by the arithmetic mean of all measurements from that probe, then averaging the result. Protocol for the procedures described above is available at Protocols.io (https://doi.org/10.17504/protocols.io.3byl4zkwjvo5/v1).

## Nuclear pore analyses

Nuclear pores were initially chosen by defining the approximate 3D coordinates of each pore in the raw dataset. A cuboidal region surrounding each coordinate (300 nm on each axis) was then extracted from the raw (Fig. S1 F), VE (Fig. 3 N), AIE (Fig. 3 O), and AIVE (Fig. 3 P) image stacks. These pores were then arranged into an identical orientation as follows: All four representations of data (raw, VE, AIE, and AIVE) for each pore were averaged to create datasets for inter-pore alignment, without biasing toward any particular method. Using the Fijiyama plugin (V4.0.11) for FIJI (v1.53), each averaged pore was manually aligned to the next pore in the sequence to generate a preliminary alignment between all 18 pores, which was then refined via automatic block matching, also through Fijiyama (Fernandez and Moisy, 2021). Individual transform operations describing this alignment were then composed into one rigid transformation per nuclear pore, allowing each pore to be aligned into the same orientation across all four processing methods. Feret diameters of the pores were measured by binarizing the pore and then applying the built-in "analyze particles" function in FIJI (v1.53) at the centrally aligned slice of the pore. All depictions of the individual nuclear pores are shown after alignment (Fig. 3, N–P and Fig. S1 F) to account for the variable orientation of some pores in the original dataset (Fig. S1 E). Procedure for the alignments described above is available at Protocols.io (https://doi.org/10.17504/protocols.io.14egn4zwqv5d/v1).

## 3D distance measurements, morphometric analyses, and rendering

All 3D surfaces were generated using the marching cubes algorithm with octree binning (256 points per leaf) in ParaView (v5.7; RRID:SCR_002516). All separation distance measurements were conducted on a surface-to-surface basis, which differs from the voxel-based distance thresholds used in earlier iterations of AIVE (Lee et al., 2024; Nguyen et al., 2021). Distance measurements and mapping for visualization were conducted in ParaView by calculating the signed distance field for one

polygonal mesh then measuring the field value at each vertex of a second polygonal mesh. The signed distance fields were calculated using a custom ParaView filter proposed in a blog post by Cory Quammen (technical leader in the Scientific Computing Team; Kitware); the custom ParaView filter has been provided on GitHub (https://github.com/BenPadman/AIVE.git). All 3D renderings were generated using the Cycles render engine in Blender (v2.93; RRID:SCR_008606). All other morphometric analyses were conducted using the MorphoLibJ plugin for FIJI (Legland et al., 2016). Differences in voxel values ($\Delta V$) were calculated using the built-in "image calculator" function in FIJI (v1.53). Mathematica (version 9) was used for all other statistical analyses and charts, including the Buffon's needle simulation in Fig. 8. Procedure for distance mapping and rendering are available at Protocols.io (https://doi.org/10.17504/protocols.io.dm6gpdrz8gzp/v1).

### Statistical analyses

Summary statistics are reported for all data as specified in the respective figure legends. Results are primarily descriptive, and thus statistical hypothesis testing was not used in any of the numerical analyses conducted. To prevent the dichotomization of results, they should instead be interpreted as a continuum (McShane et al., 2019). Readers are instructed to critically assess the magnitude, direction, and precision of all effects reported.

### Online supplemental material

Fig. S1 shows the supplementary AIVE descriptions and demonstrations. Fig. S2 shows the data at different stages of AIVE processing. Fig. S3 shows that the AIVE can also function in 2D. Fig. S4 shows the AIVE of mouse skeletal muscle. Fig. S5 shows the simulated intrusive contacts. Fig. S6 shows the additional examples of intrusive contact sites. Video 1 shows the general membrane distance probes. Video 2 shows the mitochondrial membrane distance probes. Video 3 shows the overview cutaways. Video 4 shows the smallest and largest mitochondria. Video 5 shows the most elongated and longest mitochondria. Video 6 shows the spherical, nanotunneling, and unremarkable mitochondria. Video 7 shows the AIVE of mouse skeletal muscle. Video 8 shows the intrusive contact by ER. Video 9 shows the intrusive contact by mitochondria.

### Data availability

The EM data have been deposited with annotations on Electron Microscopy Public Image Archive (RRID:SCR_019237). The code required for AIVE is available on GitHub, as individual scripts for ImageJ/FIJI (github.com/BenPadman/AIVE) (https://doi.org/10.5281/zenodo.15429332), and as a compiled ImageJ/FIJI plugin incorporating all scripts (github/BenPadman/AIVE/tree/Fiji-plugin). User guides are available within the plugin, and protocols are available on protocols.io (https://doi.org/10.17504/protocols.io.14egn4zwqv5d/v1, https://doi.org/10.17504/protocols.io.3byl4zkwjvo5/v1, https://doi.org/10.17504/protocols.io.dm6gpdrz8gzp/v1, https://doi.org/10.17504/protocols.io.14egn48x6v5d/v1, and https://doi.org/10.17504/protocols.io.36wgq691klk5/v1). An earlier version of this manuscript was posted to bioRxiv on 21st November 2024 (https://doi.

org/10.1101/2024.11.20.624606). The data, code, protocols, and key lab materials used and generated in this study are listed in a Key Resource Table alongside their persistent identifiers at (https://doi.org/10.5281/zenodo.16259276).

## Acknowledgments

We acknowledge Gediminas Gervinskas and the Ramaciotti Centre for Cryo-Electron Microscopy, Monash University, for technical support with EM. And we thank Matthew Eramo and Mike Ryan for the mouse tissue FIB-SEM dataset and acknowledge Monash Animal Research Platform, Monash University. We also acknowledge the scientific and technical assistance of Microscopy Australia at the Centre for Microscopy, Characterisation & Analysis, The University of Western Australia.

The study is funded by the joint efforts of The Michael J. Fox Foundation for Parkinson's Research (MJFF) and the Aligning Science Across Parkinson's (ASAP) initiative. MJFF administers the grant ASAP-000350 (to M. Lazarou) on behalf of ASAP and itself. National Health and Medical Research Council (GNT1106471 and GNT2033297 to M. Lazarou), the Australian Research Council Discovery Project (DP200100347 to M. Lazarou) and the Rebecca Cooper Foundation Fellowship (RC20241396 to M. Lazarou)

Author contributions: B.S. Padman: conceptualization, data curation, formal analysis, investigation, methodology, project administration, resources, software, validation, visualization, and writing—original draft, review, and editing. R.S.J. Lindblom: data curation, formal analysis, investigation, methodology, project administration, resources, software, validation, visualization, and writing—review and editing. M. Lazarou: conceptualization, funding acquisition, methodology, project administration, resources, supervision, validation, and writing—original draft, review, and editing.

Disclosures: All authors have completed and submitted the ICMJE Form for Disclosure of Potential Conflicts of Interest. M. Lazarou reported other from Automera outside the submitted work. No other disclosures were reported.

Submitted: 22 November 2024

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

 JCB

# Supplemental material

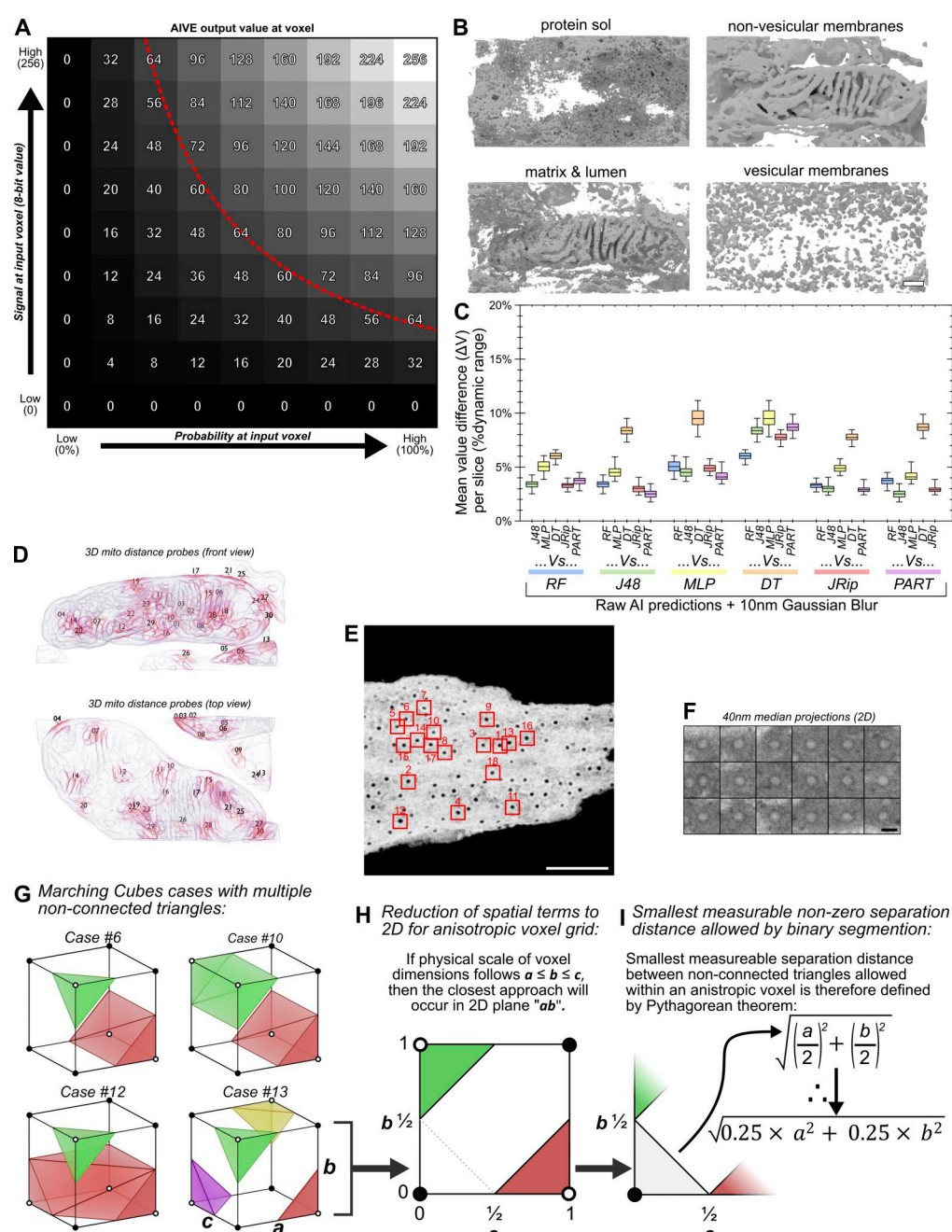

**Figure S1. Supplementary AIVE descriptions and demonstrations. (A)** Signal matrix depicting how the input values for a voxel contribute to the output value during AIVE (red line indicates approximate location of threshold used for 3D rendering and segmentation). **(B)** 3D examples of other various categories of electron signal processed via AIVE, as detected in the test dataset shown in Fig. 1. **(C)** Quantification of the average ΔV per slice for each two-way comparison made between the model predictions from Fig. 1 F after the application of a 10-nm Gaussian blur (3D renders of corresponding data also provided in Fig. S2 B). **(D)** 3D renderings indicating the position of each static 3D probe used for mitochondrial distance measurements in Fig. 3, viewed from the front and above (membranes nearest to each 3D probe are indicated by red shading). See Video 2 for rotation animation showing probe positions. **(E)** Maximum value projection of AIVE-processed data for the nucleus used in Fig. 3, depicting the position of each nuclear pore used for analysis in Fig. 3, N–T. **(F)** Averaged value projections of the unprocessed EM data after spatial alignment for analyses shown in Fig. 3, N–T. **(G–I)** A mathematical description of the smallest nonzero distance that can be measured between any two objects segmented by conventional binarization methods, after 3D reconstruction via the marching cubes algorithm. **(G)** Unique cases of cubes triangulated via the marching cubes algorithm with multiple non-connected triangles (Lorensen and Cline, 1987), which is the minimum requirement for representing two separate objects (cases 3, 4, and 7 omitted for brevity). Circles at the cube corners represent the eight voxel values used by the marching cubes algorithm to triangulate surfaces in each case. For binarized input data, the triangulated surface vertices are generated at the exact mid-point between voxel centers, which are equidistant if the voxels are spatially isometric. **(H)** If the spatial scale of one voxel dimension ("c") exceeds the other two ("a" and "b"), then the separation distance between vertices in that dimension will always be greater; by extension, the shortest nonzero distance between surfaces cannot occur in that dimension, so it can be ignored. **(I)** The shortest possible measureable nonzero distance between two binarized objects in an anisotropic voxel grid can therefore be determined by the Pythagorean theorem, in the 2D plane defined by dimensions a and b. Interquartile range for chart is indicated by box, median is indicated by horizontal lines, minimum and maximum are indicated by whiskers. Scale bars; B, 200 nm; E, 1,000 nm; F, 100 nm.

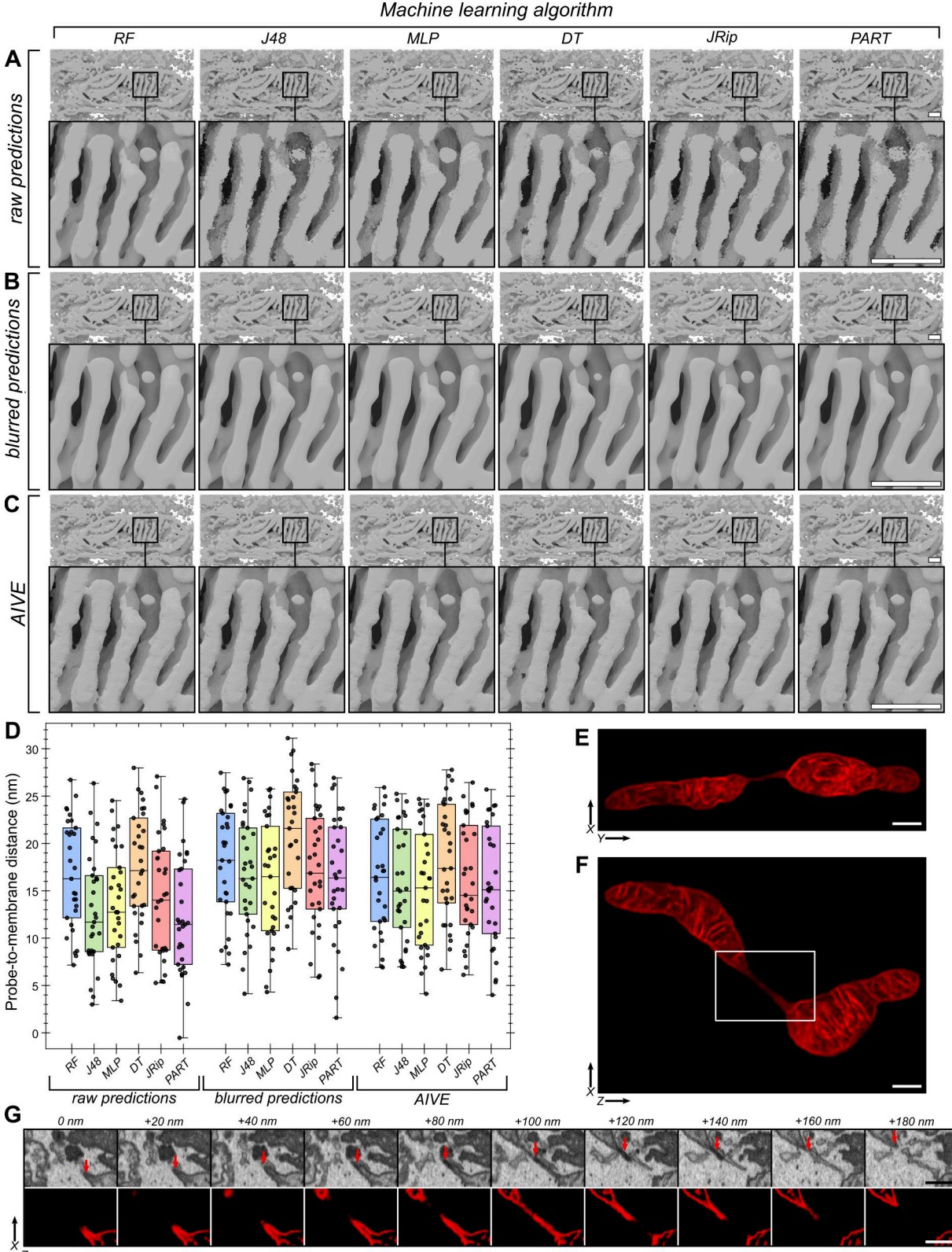

Figure S2. **Data at different stages of AIVE processing. (A–C)** 3D depictions of 2D data shown in Figs. 1 and 2, showing the 3D surfaces generated using (A) raw predictions, (B) blurred predictions, or (C) AIVE with one of six different trained models (RF, J48, MLP, DT, JRip, and PART). **(D)** Raw distance measurements between the 3D probes (shown in Fig. 2 H) and membrane surfaces under every condition shown in A–C. **(E–G)** Raw AIVE data from mitochondrion #33 (set 1) displayed as a sum projection from (E) the front and (F) above (also see Video 6), with (G) an orthoslice montage of EM and AIVE data through the mitochondrial nanotunnel (position shown by box in F). Scale bars; A–C, 200 nm; E and F, 500 nm. MLP, multilayer perceptron; DT, decision table; PART, projective adaptive resonance theory.

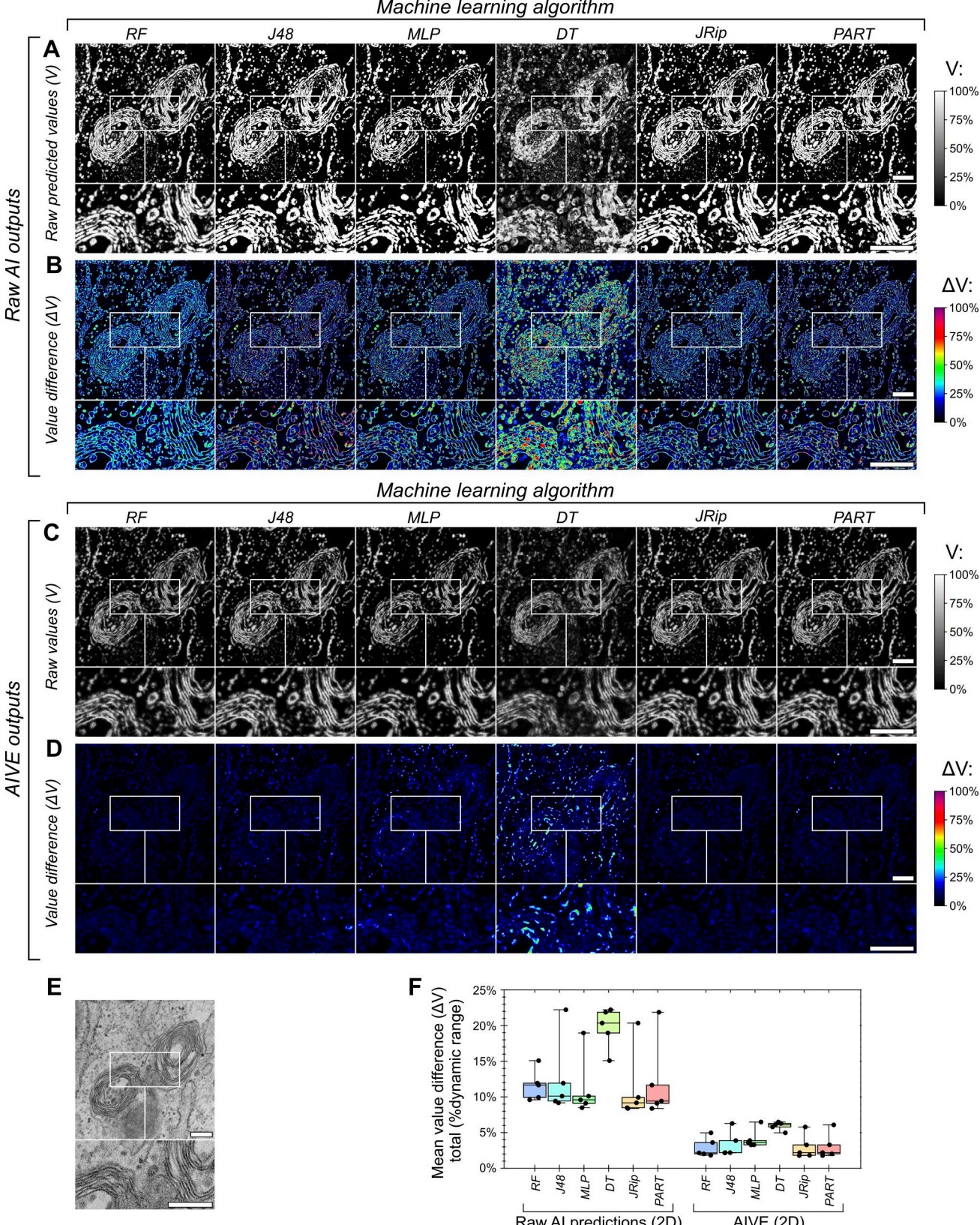

Figure S3.  **AIVE can also function in 2D. (A–E)** Raw membrane predictions and (C and D) AIVE results for (E) one TEM image, made using six different machine learning algorithms; RF, J48, MLP (multilayer perceptron), DT (decision table), JRip, and PART (projective adaptive resonance theory). **(B and D)** Visual depiction of the absolute difference in voxel values (ΔV) between each result from A and C, showing the average value difference for each two-way value comparison as a percentage of the total dynamic range (0–1). **(F)** Quantification of the average ΔV in the images for each two-way comparison between each of the (A) model predictions and (B) each of the AIVE outputs generated using those same six models. Scale bars; A–E, 200 nm.

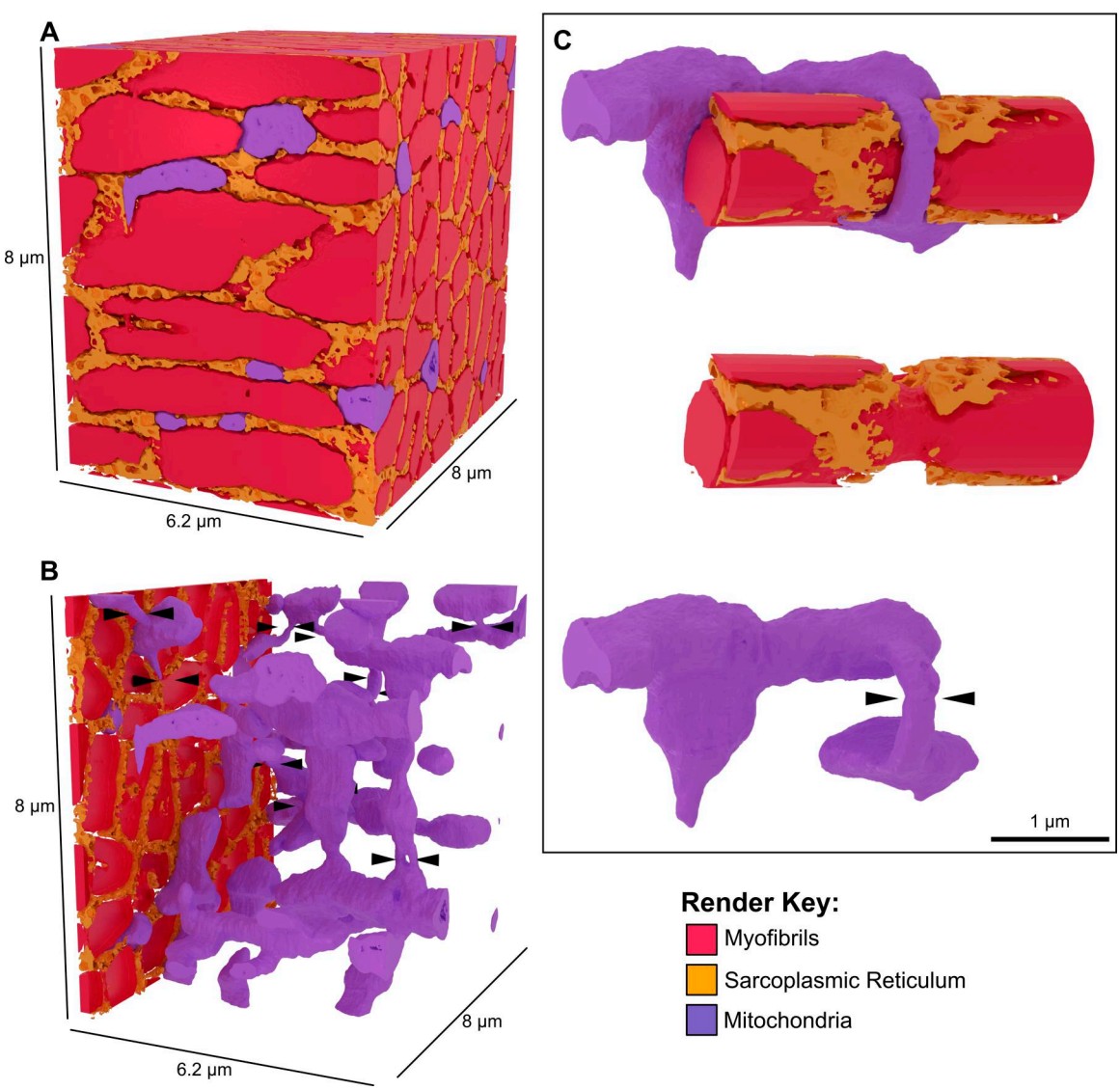

Figure S4. **AIVE of mouse skeletal muscle. (A–C)** AIVE reconstruction of mouse skeletal muscle tissue revealing mitochondrial nanotunnel connections. **(A)** 3D rendering depicting myofibrils (red), mitochondria (purple), and sarcoplasmic reticulum (gold) within a volume measuring 8 × 8 × 6.2 µm FIB-SEM volume, with (B) a rendered cutaway displaying the interconnectivity of mitochondria within the tissue (mitochondrial nanotunnels indicated by black arrowheads). **(C)** Representative example of one nanotunnel linking the mitochondrial network around a myofibril. See Video 7 for an animated depiction of data in A–C. Scale bars; A–C, as indicated within figure.

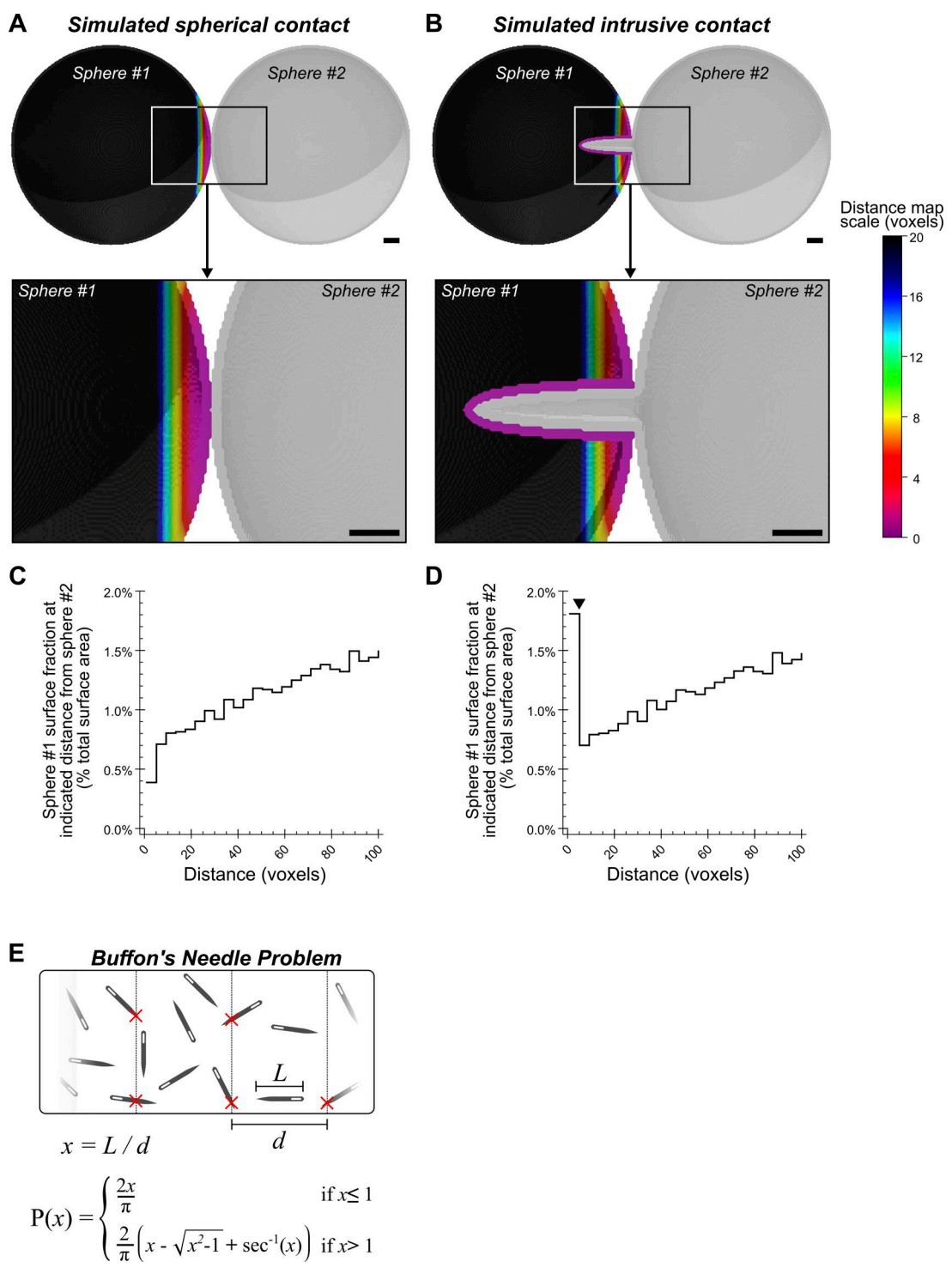

**A** *Simulated spherical contact*

Sphere #1   Sphere #2

Sphere #1   Sphere #2

**B** *Simulated intrusive contact*

Sphere #1   Sphere #2

Sphere #1   Sphere #2

Distance map scale (voxels)

**C**

**D**

**E** *Buffon's Needle Problem*

$$x = L / d$$

$$P(x) = \begin{cases} \dfrac{2x}{\pi} & \text{if } x \leq 1 \\ \dfrac{2}{\pi}\left(x - \sqrt{x^2-1} + \sec^{-1}(x)\right) & \text{if } x > 1 \end{cases}$$

Figure S5. **Simulated intrusive contacts. (A and B)** 3D renderings of simulated contact between two spheres (128 voxel diameter) with a colorimetric distance scale indicating the distance of sphere #1 from sphere #2, (A) when both structures are simple spheres, or (B) when sphere #2 intrudes into Sphere #1. **(C and D)** Surface area histograms depicting the surface area percentage of sphere #1 that is located within the indicated distances of sphere #2, (C) when both structures are simple spheres, or (D) when sphere #2 intrudes into sphere #1 (arrowhead in D indicates the anomalous peak discovered in Fig. 6). **(E)** Illustrative description of "Buffon's Needle Problem" with corresponding equations, which describe the probability of a randomly distributed needle of known length ("L") overlapping a line if those lines are regularly spaced by a known distance (d). Histogram data in C and D are surface area percentages binned into five voxel distance brackets relative to sphere #2. Scale bars in A and B, 10 voxels.

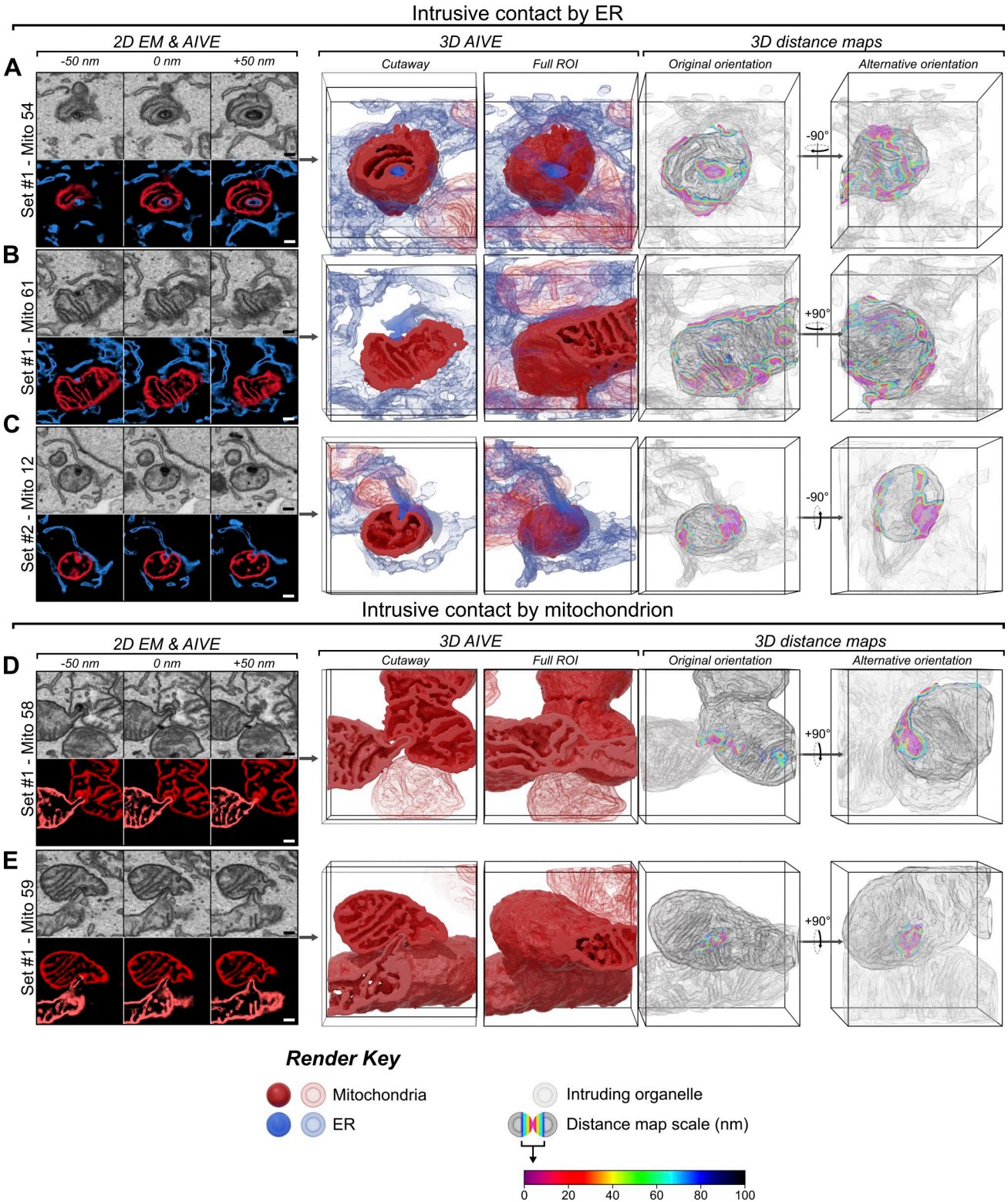

Figure S6. **Additional examples of intrusive contact sites. (A–C)** Additional examples of ER-to-mitochondrion intrusion sites, displayed as partial (50 nm) 2D-averaged projections of raw EM and AIVE data (left panels), rendered in 3D (center panels), and with a colorimetric distance map relative to the ER (right panels). **(D and E)** Additional examples of mitochondrion-to-mitochondrion intrusion sites, displayed as partial (50 nm) 2D-averaged projections of raw EM and AIVE data (left panels), rendered in 3D (center panels), and with a colorimetric distance map relative to other mitochondria (right panels). Scale bars, 200 nm; bounding dimensions of each 3D dataset frame, 1.5 × 1.5 × 1.5 µm.

Video 1.    **General membrane distance probes.** Animated rotation of the test volume, depicting the 3D membrane probe positions (from Fig. 2 H) used for membrane distance measurements in Figs. 2 and 3. General membranes are shown in diffuse navy blue, while membranes nearest to a 3D probe are indicated by red shading.

Video 2.    **Mitochondrial membrane distance probes.** Animated rotation of the test volume, depicting the 3D mitochondrial membrane probe positions (from Fig. S1 D) used for mitochondrial membrane distance measurements in Fig. 4. Mitochondrial membranes are shown in diffuse navy blue, while membranes nearest to a 3D probe are indicated by red shading.

Video 3.    **Overview cutaways.** Animated cutaways of the data presented in Fig. 4, B–D. The video begins with all organelles shown before cutting away the ER and nuclear envelope (Nu), followed by the Golgi and vesicles (VE), and finally, the EEs, LEs, and LDs. The video concludes by revealing these organelles in reverse order. Render key is provided within the animation.

Video 4.    **Smallest and largest mitochondria.** Animation depicting the structure of the smallest (Fig. 5 A) and largest (Fig. 5 B) volume mitochondria from Fig. 5. Render key is provided within the animation.

Video 5.    **Most elongated and longest mitochondria.** Animation depicting the structure of the most elongated (Fig. 5 D) and longest (Fig. 5 E) mitochondria from Fig. 5. Render key is provided within the animation.

Video 6.    **Spherical, nanotunneling, and unremarkable mitochondria.** Animation depicting the structure of the most spherical mitochondrion (Fig. 5 F), a nanotunneling mitochondrion (Fig. 5 C), and the "least remarkable" mitochondrion (Fig. 5 G) from Fig. 5. Render key is provided within the animation.

Video 7.    **AIVE of mouse skeletal muscle.** Animation of rendered data shown in Fig. S6, displaying an AIVE reconstruction of FIB-SEM data from mouse skeletal muscle tissue, measuring 8 × 8 × 6.2 µm. As the video begins, myofibrils (red) and sarcoplasmic reticulum (gold) are cut away to reveal the mitochondrial network (purple). Black arrowheads then appear to highlight the mitochondrial nanotunnels as the camera pans around the dataset. The video concludes by focusing on a specific mitochondrial nanotunnel, which is tightly wrapped around a myofibril, thus connecting mitochondria on opposing sides of the myofibril.

Video 8.    **Intrusive contact by ER.** Animation of the ER intrusions (Fig. 7, A–C) depicted in Fig. 7. The video begins with the full ROI centered around the intrusion sites, with colorations indicated by render key. As the intruding ER fades away, the mitochondrial surface is transitioned to the distance map (0–100 nm) color scale shown in Fig. 7 before rotation of the camera vertically over the datasets. The video concludes by returning to the initial shading.

Video 9.    **Intrusive contact by mitochondria.** Animation of the mitochondrial intrusions (Fig. 7, D and E) depicted in Fig. 7. The video begins with the full ROI centered around the intrusion sites, with colorations indicated by render key. As the intruding mitochondrion fades away, the mitochondrial surfaces are transitioned to the distance map (0–100 nm) color scale shown in Fig. 7 before rotation of the camera vertically over the datasets. The video concludes by returning to the initial shading.

