## [Peer Review File · The Journal of Cell Biology]

AI-directed voxel extraction and volume EM identify intrusions as sites of mitochondrial contact

Benjamin Padman, Runa Lindblom, and Michael Lazarou

Corresponding Author(s): Michael Lazarou, Walter and Eliza Hall Institute of Medical Research and Michael Lazarou, Walter and Eliza Hall Institute of Medical Research

Review Timeline:

Submission Date:	2024-11-22
Editorial Decision:	2025-01-14
Revision Received:	2025-05-16
Editorial Decision:	2025-06-26
Revision Received:	2025-07-03

Monitoring Editor: Eva Nogales

Scientific Editor: Dan Simon

Transaction Report:

DOI: <https://doi.org/10.1083/jcb.202411138>

January 14, 2025

Re: JCB manuscript #202411138

Michael Lazarou
Walter and Eliza Hall Institute of Medical Research

Dear Dr. Lazarou,

Thank you for submitting your manuscript entitled "AI-directed voxel extraction and volume EM identify intrusions as sites of mitochondrial contact." Your manuscript has been assessed by expert reviewers, whose comments are appended below. Although the reviewers express potential interest in this work, significant concerns unfortunately preclude publication of the current version of the manuscript in JCB.

You will see that Reviewer #1 is enthusiastic about the AIVE workflow and asks for more details about statistical analysis, to clarify how distance measurements were done, demonstrate more directly that the newly identified mitochondrial intrusions would have been missed by traditional approaches, and has a few discussion points. Reviewer #2 while also supportive of AIVE, raises more substantial conceptual concerns. This reviewer asks whether AIVE works better than two similar methods that were not cited in your paper. Demonstrating that a new method or workflow performs better than existing approaches is required for the JCB Tools format so this must be addressed, preferably by direct comparison rather than just discussing differences between the workflows. Reviewer #2 also notes two papers that have already used AIVE so you should clearly explain why these previous usages of AIVE did not fully describe the method and also note any new features. A Tools paper would not be expected to provide a substantial amount of functional characterization but if you can provide any data regarding the nature of the mitochondrial intrusions, that would certainly enhance the impact of the study.

Please let us know if you are able to address the major issues outlined above and wish to submit a revised manuscript to JCB. Note that a substantial amount of additional data likely would be needed to satisfactorily address the concerns of the reviewers. The typical timeframe for revisions is three to four months. If you anticipate any difficulties in meeting this aforementioned revision time limit, please contact us and we can work with you to find an appropriate time frame for resubmission. Please note that papers are generally considered through only one revision cycle, so any revised manuscript will likely be either accepted or rejected.

If you choose to revise and resubmit your manuscript, please also attend to the following editorial points. Please direct any editorial questions to the journal office.

GENERAL GUIDELINES:

Text limits: Character count is < 40,000, not including spaces. Count includes title page, abstract, introduction, results, discussion, and acknowledgments. Count does not include materials and methods, figure legends, references, tables, or supplemental legends.

Figures: Your manuscript may have up to 10 main text figures. To avoid delays in production, figures must be prepared according to the policies outlined in our Instructions to Authors, under Data Presentation, <https://jcb.rupress.org/site/misc/ifora.xhtml>. All figures in accepted manuscripts will be screened prior to publication.

Supplemental information: There are strict limits on the allowable amount of supplemental data. Your manuscript may have up to 5 supplemental figures. Up to 10 supplemental videos or flash animations are allowed. A summary of all supplemental material should appear at the end of the Materials and methods section.

Please note that JCB now requires authors to submit Source Data used to generate figures containing gels and Western blots with all revised manuscripts. This Source Data consists of fully uncropped and unprocessed images for each gel/blot displayed in the main and supplemental figures. If your revised paper will include cropped gel and/or blot images, please be sure to provide one Source Data file for each figure that contains gels and/or blots along with your revised manuscript files. File names for Source Data figures should be alphanumeric without any spaces or special characters (i.e., SourceDataF#, where F# refers to the associated main figure number or SourceDataFS# for those associated with Supplementary figures). The lanes of the gels/blots should be labeled as they are in the associated figure, the place where cropping was applied should be marked (with a box), and molecular weight/size standards should be labeled wherever possible. Source Data files will be made available to reviewers during evaluation of revised manuscripts and, if your paper is eventually published in JCB, the files will be directly

linked to specific figures in the published article.

If you choose to resubmit, please include a cover letter addressing the reviewers' comments point by point. Please also highlight all changes in the text of the manuscript.

Regardless of how you choose to proceed, we hope that the comments below will prove constructive as your work progresses. We would be happy to discuss them further once you've had a chance to consider the points raised. You can contact the journal office with any questions at cellbio@rockefeller.edu.

Thank you for thinking of JCB as an appropriate place to publish your work.

Sincerely,

Eva Nogales, PhD
Monitoring Editor
Journal of Cell Biology

Dan Simon, PhD
Scientific Editor
Journal of Cell Biology

Reviewer #1 (Comments to the Authors (Required)):

1. This paper aims to develop a computational strategy to reliably and accurately segment organelle boundaries. These regions are notoriously miscalculated when using artificial intelligence (AI)-based methods for semantic segmentation of membranes from focused ion beam-scanning electron microscopy (FIB-SEM) imaging data. Their signal masking strategy (AIVE) uses the electron signal from the FIB-SEM image to 'trim' the AI-predicted segmentation such that only the regions that overlap with the electron signal are included in the final segmentation. Their method significantly reduces the value difference across all benchmarked AI-assisted segmentation outputs. They further demonstrate that their method improves many other aspects of AI-assisted segmentation output, including organelle classification, organelle distance measurements, and the segmentation accuracy of large macromolecular complexes (i.e., nuclear pore complex). They demonstrate that their method can deconvolute the complexity of mitochondrial shapes and morphologies within the entire network, and they can use skeletonization of their segmentations to classify mitochondria into different shapes automatically. They then demonstrate proof-of-concept that their method can discover new biological insights, focusing on membrane contact sites since these are the primary feature that would otherwise suffer from incorrect boundary segmentation. They discover a distinct type of contact site between the endoplasmic reticulum and the mitochondria-one they term an 'intrusive' contact. This contact is characterized by a close inter-organelle distance (less than 5 nm) that covers a large surface area. They use modeling to determine that previous reports using traditional EM imaging have likely missed these intrusion contacts due to the low probability of capturing these contacts in the thin sections required for traditional EM imaging.

This manuscript presents a helpful strategy for improving semantic segmentations from AI programs. The data presented throughout the manuscript provides a convincing argument that their method should be incorporated into FIB-SEM processing workflows to achieve more accurate segmentations that can lead to new biological insights. Given that the scripts are publicly available, I anticipate they will likely be used by many groups who routinely process FIB-SEM data using AI-assisted approaches.

2. Overall, I am impressed with the degree of benchmarking the authors perform to show that their method produces superior results. Furthermore, demonstrating that their approach leads to new biological insights proves that this should become a standardized part of the FIB-SEM workflow. All of the paper's main points are associated with convincing data to support the claim.

3. A few minor suggestions would improve the overall clarity of the manuscript:

1. The authors should provide more detail regarding the statistical analyses used throughout the manuscript and justification for why these tests are appropriate.

2. It was unclear whether the distance quantifications were generated from a voxel-to-voxel distance or nearest-neighbor calculations between triangles in a surface mesh. Some clarification of this would be useful to appreciate the accuracy of their

measurements below 5 nm.

3. The authors provide a nice justification for why historical TEM imaging may have missed these intrusive contacts. However, they do not demonstrate that these contacts would have been missed without their AIVE approach. While extensive benchmarking is performed in the first part of the manuscript, I recommend the authors demonstrate the degree to which their approach sheds light on these contacts that would have otherwise been missed if only AI-assisted segmentation were performed.

4. It would be helpful if the authors could comment on whether their approach would work for AI-generated semantic segmentation derived from data from other sources, such as TEM.

5. The authors present their strategy as an additional step downstream of AI-generated segmentation. Some discussion of the future potential of streamlining both the process of segmentation and this clean-up step may be helpful to the broader community of developers designing new AI-assisted strategies.

Reviewer #2 (Comments to the Authors (Required)):

Padman et al. report a method for analyzing volumetric FIB-SEM data they call AIVE (AI-directed Voxel Extraction), in which they scale AI predictions with raw data (voxel values) for improved segmentation. Specifically, AI outputs (e.g., membrane segmentation probabilities from ML models) are multiplied with normalized voxel intensity values from the original EM data. This step acts as a signal "mask" to rescale uncertain AI outputs using the actual data, enhancing segmentation reliability. The authors demonstrate that this improves precision using credible benchmark tests. However, I am unsure whether the method is sufficiently novel to warrant publication in JCB, as it seems similar to prior work. Would the authors include a more thorough review of precedents to help the reader distinguish their method, especially from papers like these:

Domain adversarial networks and intensity-based data augmentation for male pelvic organ segmentation in cone beam CT; by Brion et al. in *Computers in Biology and Medicine*, 2021.

CleftNet: Augmented deep learning for synaptic cleft detection from brain electron microscopy; by Y. Liu et al. from *IEEE Transactions on Medical Imaging*, 2021.

The authors also cite [16, 20] as what appear to be prior uses of AIVE:

16. Nguyen, T.N., et al., ATG4 family proteins drive phagophore growth independently of the LC3/GABARAP lipidation system, from *Mol Cell*, 2021. 81(9): p. 2013-2030 e9.

20. Lee, R.G., et al., Quantitative subcellular reconstruction reveals a lipid mediated inter-organelle biogenesis network. from *Nat Cell Biol*, 2024. 26(1): p. 57-71.

Please clarify which aspects of their method were insufficiently reported in these manuscripts.

Finally, while the intrusions of mitochondria into other mitochondria and the ER into mitochondria are curious observations and excellent examples of the power of volumetric FIB-SEM, their observation alone without some degree of functional characterization may not rise to the level of a JCB manuscript.

Reviewer #1 (Comments to the Authors (Required)):

1. This paper aims to develop a computational strategy to reliably and accurately segment organelle boundaries. These regions are notoriously miscalculated when using artificial intelligence (AI)-based methods for semantic segmentation of membranes from focused ion beam-scanning electron microscopy (FIB-SEM) imaging data. Their signal masking strategy (AIVE) uses the electron signal from the FIB-SEM image to 'trim' the AI-predicted segmentation such that only the regions that overlap with the electron signal are included in the final segmentation. Their method significantly reduces the value difference across all benchmarked AI-assisted segmentation outputs. They further demonstrate that their method improves many other aspects of AI-assisted segmentation output, including organelle classification, organelle distance measurements, and the segmentation accuracy of large macromolecular complexes (i.e., nuclear pore complex). They demonstrate that their method can deconvolute the complexity of mitochondrial shapes and morphologies within the entire network, and they can use skeletonization of their segmentations to classify mitochondria into different shapes automatically. They then demonstrate proof-of-concept that their method can discover new biological insights, focusing on membrane contact sites since these are the primary feature that would otherwise suffer from incorrect boundary segmentation. They discover a distinct type of contact site between the endoplasmic reticulum and the mitochondria-one they term an 'intrusive' contact. This contact is characterized by a close inter-organelle distance (less than 5 nm) that covers a large surface area. They use modeling to determine that previous reports using traditional EM imaging have likely missed these intrusion contacts due to the low probability of capturing these contacts in the thin sections required for traditional EM imaging.

This manuscript presents a helpful strategy for improving semantic segmentations from AI programs. The data presented throughout the manuscript provides a convincing argument that their method should be incorporated into FIB-SEM processing workflows to achieve more accurate segmentations that can lead to new biological insights. Given that the scripts are publicly available, I anticipate they will likely be used by many groups who routinely process FIB-SEM data using AI-assisted approaches.

2. Overall, I am impressed with the degree of benchmarking the authors perform to show that their method produces superior results. Furthermore, demonstrating that their approach leads to new biological insights proves that this should become a standardized part of the FIB-SEM workflow. All of the paper's main points are associated with convincing data to support the claim.

We thank the reviewer for their supportive comments and their insightful and constructive feedback.

3. A few minor suggestions would improve the overall clarity of the manuscript:

1. The authors should provide more detail regarding the statistical analyses used throughout the manuscript and justification for why these tests are appropriate.

We apologize for the lack of clarity. Statistical hypothesis testing with P-values were not used in the study. Briefly, our rationale was that the analyses were either benchmarking measures of AIVE or measurements of organelle contacts that were not testing biological hypotheses i.e descriptive, rather than inferential statistics. Our aim was for readers to critically evaluate the benchmarking improvements of AIVE on a continuum rather than a dichotomous significant/not significant conclusion.

This point has now been clarified in the methods under a new heading:

“Statistical analyses

Summary statistics are reported for all data as specified in the respective figure legends. Results are primarily descriptive, and thus statistical hypothesis testing was not used in any of the numerical analyses conducted. To prevent the dichotomization of results they should instead be interpreted as a continuum [70]. Readers are instructed to critically assess the magnitude, direction, and precision of all effects reported.”

2.It was unclear whether the distance quantifications were generated from a voxel-to-voxel distance or nearest-neighbor calculations between triangles in a surface mesh. Some clarification of this would be useful to appreciate the accuracy of their measurements below 5 nm.

This an astute point. To address the reviewer's comment and improve clarity we have renamed the corresponding sub-section of the to “3D distance measurements, morphometric analyses, and rendering”. In addition, the 3D distance measurements were indeed conducted using “nearest-neighbour calculations between triangles in a surface mesh”.

To further clarify the expected accuracy & precision of <5nm distance measurements, we have now also provided a mathematical definition of the smallest non-zero distance that can be measured between any two binary segmented objects after 3D reconstruction. The new data are provided as a supplemental figure (Fig.S1G-I), with new text in the results where the “terracing artifact” is explained:

“... The terracing is caused by abrupt (binary) changes in voxel value, which affect the local shape since contours can only be represented by a limited pool of surface topologies (2^8 ; 256) [24], of which only 15 are unique [25]. The rigid pool of surface topologies imposes a limit on the smallest non-zero distance that can be measured between any two objects, which would be 2.3nm for our data (see Fig.S1G-I). In contrast, the additional information present within 8-bit scalar datasets, such as those generated by AIE, VE and AIVE (Fig. 3H-J) allows surface vertices to be positioned at any point between voxel centres, theoretically allowing for $>1.8 \times 10^{19}$ (256^8) different unique polygon configurations. These polygon configurations can also adopt fractional angles that have spatial anisotropy with the voxel grid [26], thus removing the limit on the smallest measurements that can be made between objects...”

3.The authors provide a nice justification for why historical TEM imaging may have missed these intrusive contacts. However, they do not demonstrate that these contacts would have been missed without their AIVE approach. While extensive benchmarking is performed in the first part of the manuscript, I recommend the authors demonstrate the degree to which their approach sheds light on these contacts that would have otherwise been missed if only AI-assisted segmentation were performed.

We discovered that mitochondrial intrusions were a form of contact when the separation distance data from our AIVE analyses were collated and measured (Fig 6K and L). The characteristic peak of intrusion contacts within the 0-5 nm range existed across multiple datasets and was amongst the largest proportion of contact relative to mitochondrial surface area. These measurements demonstrated to us that the contacts were beyond incidental and instead represented reproducible biological events. The benchmarking of different AI models in Fig.2 showed measurement variability of >5nm when AI was used alone versus <5nm when AIVE was applied. Given this, the boundary confidence provided by AIVE enabled the measurement and proportion of intrusions, whereas in its absence using AI alone the intrusions would have either been dismissed as being within the range of error or the characteristic intrusion peak might have been absent all together. For these reasons, it is challenging to quantitatively measure the degree that AIVE enabled the identification of intrusions beyond the fact that it enabled us to confidently measure the incidence and proportion of 0-5 nm contacts across datasets leading to their identification as mitochondrial intrusions.

The following text has been added to the results section of the manuscript to clarify this point:

" Importantly, this characteristic peak between 0 and 5 nm was unlikely be detected via conventional AI-based binary segmentation alone, since binarized data could not have measured distances between 0 and 2.3 nm (Fig.S1G-I), and the variation in AI-defined boundaries greatly exceeded 5 nm (Fig.2 I). The precision conferred by AIVE was therefore essential to the discovery of mitochondrial intrusions as a form of membrane contact."

4.It would be helpful if the authors could comment on whether their approach would work for AI-generated semantic segmentation derived from data from other sources, such as TEM.

We thank the reviewer for the insightful comment. Indeed, the principles of AIVE function in 2D just as well as they do in 3D.

We now include new data (Fig.S3) and text in the results section to demonstrate this:

"We also note that the principles of AIVE are applicable to 2D imaging methods such as Transmission Electron Microscopy (TEM). Similar to the 3D AIVE analyses (Fig. 1-2), 2D application of AIVE reduced the variability between results generated by different AI models (Fig. S3)."

An additional demonstration of this is also provided in response to reviewer 2, comment 1, where we apply AIVE to data extracted from a published figure.

5.The authors present their strategy as an additional step downstream of AI-generated segmentation. Some discussion of the future potential of streamlining both the process of segmentation and this clean-up step may be helpful to the broader community of developers designing new AI-assisted strategies.

We thank the reviewer for this helpful comment. The following text has now been added to the discussion of the manuscript:

"Moving forward, it would be beneficial to integrate the process of AIVE as a default option in AI-assisted segmentation strategies. As it stands, AIVE is a separate step to enhance AI-assisted predictions, but through incorporation within segmentation algorithms AIVE can be seamlessly applied in the future."

Reviewer #2 (Comments to the Authors (Required)):

Padman et al. report a method for analyzing volumetric FIB-SEM data they call AIVE (AI-directed Voxel Extraction), in which they scale AI predictions with raw data (voxel values) for improved segmentation. Specifically, AI outputs (e.g., membrane segmentation probabilities from ML models) are multiplied with normalized voxel intensity values from the original EM data. This step acts as a signal "mask" to rescale uncertain AI outputs using the actual data, enhancing segmentation reliability. The authors demonstrate that this improves precision using credible benchmark tests.

1. However, I am unsure whether the method is sufficiently novel to warrant publication in JCB, as it seems similar to prior work. Would the authors include a more thorough review of precedents to help the reader distinguish their method, especially from papers like these:

Domain adversarial networks and intensity-based data augmentation for male pelvic organ segmentation in cone beam CT; by Brion et al. in Computers in Biology and Medicine, 2021.

CleftNet: Augmented deep learning for synaptic cleft detection from brain electron microscopy; by Y. Liu et al. from IEEE Transactions on Medical Imaging, 2021.

We thank the reviewer for raising this point, and we acknowledge that there is a clear need to distinguish our work from the prior art. Both referenced publications use remarkably similar terminology to our manuscript, yet they describe methods related to the augmentation of AI behaviour and performance. As requested by the reviewer, new text is now included in the results section to better articulate the distinguishing features of our work. A key sentence in the abstract, which initially sounded similar to the publications referenced by the reviewer, and has now been revised as follows:

"We outline a segmentation strategy termed AI-directed Voxel Extraction (AIVE), which refines segmentation results and boundary predictions derived from any AI-based method by combining those results with electron signal values.."

New text has also been provided in the results section to explicitly highlight the difference between our work and prior art:

"Unlike methods that use EM signals to augment AI performance or training [21, 22], the core stages of AIVE occur downstream of AI-based image segmentation without influencing the training or behaviour of the AI model."

We also further demonstrate that AIVE can improve analyses of data that has already been published, such as those referenced by the reviewer. We have therefore conducted a quick demonstration by applying AIVE to data extracted from "*CleftNet: Augmented Deep Learning for Synaptic Cleft Detection from Brain Electron Microscopy*" by Yi Liu & Shuiwang Ji (2021). The data (shown below) was extracted from Figure 4C of that publication, which compared results obtained via 3D U-Net, ResUnet, AttnUnet, CleftNet, and the ground truth. A sub-region needed to be used (due to inconsistent inset box positions), and we needed to assume that the mitochondrion visible in the EM data was approximately 500nm in diameter (for the purposes of the CLAHE filter). Even with these assumptions, the segmentation data was demonstrably more consistent after being processed via AIVE. This data is provided for the reviewer's reference only.

Figure Legend: (A-B) Data extracted from Fig.4 of the article by Yi Liu & Shuiwang Ji (2021), showing cropped insets of (A) EM data from the figure and (B) binary segmentation labels assigned by a 3D U-Net, ResUnet, AttUnet, CleftNet, and a Human. (C) The EM inset after CLAHE processing, under the assumption that the visible mitochondrion was approximately 500nm in diameter. (D) The 2D AIVE processed data, computed by combining the AI predictions from (B) with the normalized EM data from (C). (E) Quantification of the average ΔV in the images for each 2-way comparison between each of the (B) model predictions and (D) each of the AIVE

outputs generated using those same six models. Scale bars not provided, as they were not available in the original publication.

2. The authors also cite [16, 20] as what appear to be prior uses of AIVE:

16. Nguyen, T.N., et al., ATG4 family proteins drive phagophore growth independently of the LC3/GABARAP lipidation system, from Mol Cell, 2021. 81(9): p. 2013-2030 e9.

20. Lee, R.G., et al., Quantitative subcellular reconstruction reveals a lipid mediated inter-organelle biogenesis network. from Nat Cell Biol, 2024. 26(1): p. 57-71.

Please clarify which aspects of their method were insufficiently reported in these manuscripts.

The reviewer raises a valuable point to clarify. The current study is the first canonical description of AIVE that demonstrates its benefits to membrane boundaries and membrane contacts through extensive benchmarking analyses, along with detailed descriptions of its theory and application. Without these analyses it would be unclear why and how a cell biologist would choose to apply AIVE to their own work and what aspects of segmentation are actually improved over AI alone.

Specifically, the prior studies did not:

- Show that AI predictions at object boundaries are the most variable predictions (Fig.1).
- Introduce a new suite of tools for benchmarking segmentation methods (Fig.1-3).
- Provide any form of quantitative benchmarking (Fig. 1-3 & S1-2).
- Explain the specific contributions of each individual stage of AIVE (Fig.1-3).
- Explain the technical basis for chosen isosurface values (Fig.S1A).
- Demonstrate that AIVE is more accurate than AI alone (Fig. 3).
- Show that AIVE can also be applied to non-membrane structures (Fig.S1B).
- Demonstrate that AIVE can be applied using any AI model (Fig. 2 & S2).
- Show that object classification for AIVE can be performed by human or AI alike (Fig.3).
- Demonstrate that AIVE is equally applicable to cultured cells and tissues (Fig. S4).
- Provide scripts for automating AIVE (Methods).
- Introduce a newly developed method for quantifying membrane contact sites (Fig. 1-3, 6).

In addition, as part of the revised manuscript we have now provided AIVE as an open source software tool for integration with new and existing analysis pipelines. The complete code library is available at github.com/BenPadman/AIVE (<https://doi.org/10.5281/zenodo.15429332>), and the code may be installed as an ImageJ plugin from github.com/BenPadman/AIVE/tree/Fiji-plugin. User guides are available within the plugin and at: dx.doi.org/10.17504/protocols.io.14egn48x6v5d/v1

Links to these key resources are now provided under our Data Availability statement:

Data Availability:

The EM data has been deposited with annotations on EMPIAR (Electron Microscopy Public Image Archive; RRID:SCR_019237). The code required for AIVE is available on GitHub, as individual scripts for ImageJ/FIJI (github.com/BenPadman/AIVE) [DOI: 10.5281/zenodo.15429332], and as a compiled ImageJ/FIJI plugin incorporating all scripts (github.com/BenPadman/AIVE/tree/Fiji-plugin). User guides are available within the plugin, and protocols are available on protocols.io [DOIs: 10.17504/protocols.io.14egn4zwwqv5d/v1, 10.17504/protocols.io.3byl4zkwjvo5/v1, 10.17504/protocols.io.dm6gpdrz8gzp/v1, 10.17504/protocols.io.14egn48x6v5d/v1, and

10.17504/protocols.io.36wgg691klk5/v1]. An earlier version of this manuscript was posted to bioRxiv on 21st November 2024 (DOI: 10.1101/2024.11.20.624606).

3. Finally, while the intrusions of mitochondria into other mitochondria and the ER into mitochondria are curious observations and excellent examples of the power of volumetric FIB-SEM, their observation alone without some degree of functional characterization may not rise to the level of a JCB manuscript.

We appreciate the reviewer's comment and agree that further analyses of these newly discovered contacts is warranted. However, given that this manuscript is a tools article, further biological evaluation of the intrusions is beyond the scope of the current study.

Reference

1. Nguyen, T.N., et al., *ATG4 family proteins drive phagophore growth independently of the LC3/GABARAP lipidation system*. *Molecular Cell*, 2021. **81**(9): p. 2013-2030.e9.

June 26, 2025

RE: JCB Manuscript #202411138R

Michael Lazarou
Walter and Eliza Hall Institute of Medical Research

Dear Dr. Lazarou,

Thank you for submitting your revised manuscript entitled "AI-directed voxel extraction and volume EM identify intrusions as sites of mitochondrial contact." We would be happy to publish your paper in JCB pending final revisions necessary to meet our formatting guidelines (see details below).

A. MANUSCRIPT ORGANIZATION AND FORMATTING:

1) Text limits: Character count for Tools is < 40,000, not including spaces. Count includes title page, abstract, introduction, results, discussion, and acknowledgments. Count does not include materials and methods, figure legends, references, tables, or supplemental legends.

2) Figure formatting: Tools may have up to 10 main text figures. Please make sure all images and inset magnifications have scale bars. Also, please avoid pairing red and green for images and graphs to ensure legibility for color-blind readers. If red and green are paired for images, please ensure that the particular red and green hues used in micrographs are distinctive with any of the colorblind types. If not, please modify colors accordingly or provide separate images of the individual channels.

3) Statistical analysis: Error bars on graphic representations of numerical data must be clearly described in the figure legend. The number of independent data points (n) represented in a graph must be indicated in the legend. Please indicate whether 'n' refers to technical or biological replicates (i.e. number of analyzed cells, samples or animals, number of independent experiments). If independent experiments with multiple biological replicates have been performed, we recommend using distribution-reproducibility SuperPlots (please see Lord et al., JCB 2020) to better display the distribution of the entire dataset, and report statistics (such as means, error bars, and P values) that address the reproducibility of the findings.

Statistical methods should be explained in full in the materials and methods. For figures presenting pooled data the statistical measure should be defined in the figure legends. Please also be sure to indicate the statistical tests used in each of your experiments (both in the figure legend itself and in a separate methods section) as well as the parameters of the test (for example, if you ran a t-test, please indicate if it was one- or two-sided, etc.). Also, if you used parametric tests, please indicate if the data distribution was tested for normality (and if so, how). If not, you must state something to the effect that "Data distribution was assumed to be normal but this was not formally tested."

4) Materials and methods: Should be comprehensive and not simply reference a previous publication for details on how an experiment was performed. Please provide full descriptions (at least in brief) in the text for readers who may not have access to referenced manuscripts. The text should not refer to methods "...as previously described."

5) For all cell lines, vectors, strains, constructs/cDNAs, etc. - all genetic material: please include database / vendor ID (e.g. Addgene, ATCC, etc.) or if unavailable, please briefly describe their basic genetic features, even if described in other published work or gifted to you by other investigators (and provide references where appropriate). Please be sure to provide the sequences for all of your oligos: primers, si/shRNA, RNAi, gRNAs, etc. in the materials and methods. You must also indicate in the methods the source, species, and catalog numbers/vendor identifiers (where appropriate) for all of your antibodies, including secondary. If antibodies are not commercial, please add a reference citation if possible.

6) Microscope image acquisition: The following information must be provided about the acquisition and processing of images:

- a. Make and model of microscope
- b. Type, magnification, and numerical aperture of the objective lenses
- c. Temperature
- d. Camera make and model
- e. Acquisition software
- f. Any software used for image processing subsequent to data acquisition. Please include details and types of operations

involved (e.g., type of deconvolution, 3D reconstitutions, surface or volume rendering, gamma adjustments, etc.).

7) References: There is no limit to the number of references cited in a manuscript. References should be cited parenthetically in the text by author and year of publication. Abbreviate the names of journals according to PubMed.

8) Supplemental materials: Tools generally have up to 5 supplemental figures and 10 videos. You currently exceed this limit but, in this case, we will be able to give you the extra space. Tables, like figures, should be provided as individual, editable files. A summary of all supplemental material should appear at the end of the Materials and methods section. Please include one brief sentence per item.

9) Video legends: Should describe what is being shown, the cell type or tissue being viewed (including relevant cell treatments, concentration and duration, or transfection), the imaging method (e.g., time-lapse epifluorescence microscopy), what each color represents, how often frames were collected, the frames/second display rate, and the number of any figure that has related video stills or images.

10) eTOC summary: A ~40-50 word summary that describes the context and significance of the findings for a general readership should be included on the title page. The statement should be written in the present tense and refer to the work in the third person. It should begin with "First author name(s) et al..." to match our preferred style.

11) Conflict of interest statement: JCB requires inclusion of a statement in the acknowledgements regarding competing financial interests. If no competing financial interests exist, please include the following statement: "The authors declare no competing financial interests." If competing interests are declared, please follow your statement of these competing interests with the following statement: "The authors declare no further competing financial interests."

12) A separate author contribution section is required following the Acknowledgments in all research manuscripts. All authors should be mentioned and designated by their first and middle initials and full surnames. We encourage use of the CRediT nomenclature (<https://casrai.org/credit/>).

13) ORCID IDs: ORCID IDs are unique identifiers allowing researchers to create a record of their various scholarly contributions in a single place. Please note that ORCID IDs are required for all authors. At resubmission of your final files, please be sure to provide your ORCID ID and those of all co-authors.

14) Journal of Cell Biology now requires a data availability statement for all research article submissions. These statements will be published in the article directly above the Acknowledgments. The statement should address all data underlying the research presented in the manuscript. Please visit the JCB instructions for authors for guidelines and examples of statements at (<https://rupress.org/jcb/pages/editorial-policies#data-availability-statement>).

B. FINAL FILES:

Thank you for your attention to these final processing requirements. Please revise and format the manuscript and upload materials within 7 days. If you need an extension for whatever reason, please let us know and we can work with you to determine a suitable revision period.

Thank you for this interesting contribution, we look forward to publishing your paper in Journal of Cell Biology.

Sincerely,

Eva Nogales, PhD
Monitoring Editor
Journal of Cell Biology

Dan Simon, PhD
Scientific Editor
Journal of Cell Biology

Reviewer #1 (Comments to the Authors (Required)):

The authors have addressed my concerns and suggestions for improvement. I approve of the manuscript in its current form for publication as a Tools article in JCB.